# A post-transcriptional program coordinated by CSDE1 prevents intrinsic neural differentiation of human embryonic stem cells

Hyun Ju Lee[1], Deniz Bartsch [1,2], Cally Xiao[1,3], Santiago Guerrero[4,5], Gaurav Ahuja[1,2], Christina Schindler[1], James J. Moresco[6], John R. Yates III[6], Fátima Gebauer[4], Hisham Bazzi[1,3], Christoph Dieterich[7], Leo Kurian [1,2] & David Vilchez [1]

While the transcriptional network of human embryonic stem cells (hESCs) has been extensively studied, relatively little is known about how post-transcriptional modulations determine hESC function. RNA-binding proteins play central roles in RNA regulation, including translation and turnover. Here we show that the RNA-binding protein CSDE1 (cold shock domain containing E1) is highly expressed in hESCs to maintain their undifferentiated state and prevent default neural fate. Notably, loss of CSDE1 accelerates neural differentiation and potentiates neurogenesis. Conversely, ectopic expression of CSDE1 impairs neural differentiation. We find that CSDE1 post-transcriptionally modulates core components of multiple regulatory nodes of hESC identity, neuroectoderm commitment and neurogenesis. Among these key pro-neural/neuronal factors, CSDE1 binds fatty acid binding protein 7 (*FABP7*) and vimentin (*VIM*) mRNAs, as well as transcripts involved in neuron projection development regulating their stability and translation. Thus, our results uncover CSDE1 as a central post-transcriptional regulator of hESC identity and neurogenesis.

[1] Cologne Excellence Cluster for Cellular Stress Responses in Aging-Associated Diseases (CECAD), University of Cologne, Joseph Stelzmann Strasse 26, 50931 Cologne, Germany. [2] Laboratory for Developmental and Regenerative RNA biology, Center for Molecular Medicine Cologne (CMMC), University of Cologne, Robert-Koch-Str. 21, 50931 Cologne, Germany. [3] Department of Dermatology and Venereology, University Hospital of Cologne, Joseph Stelzmann Strasse 26, 50931 Cologne, Germany. [4] Gene Regulation, Stem Cells and Cancer Programme, Centre for Genomic Regulation (CRG), The Barcelona Institute of Science and Technology, 08003 Barcelona, Spain. [5] Centro de Investigación Genética y Genómica, Facultad de Ciencias de la Salud Eugenio Espejo, Universidad Tecnologica Equinoccial, Avenue Mariscal Sucre, 170129 Quito, Ecuador. [6] Department of Chemical Physiology, 10550 North Torrey Pines Road, SR111, The Scripps Research Institute, La Jolla, CA 92037, USA. [7] Section of Bioinformatics and Systems Cardiology, Department of Internal Medicine III and Klaus Tschira Institute for Computational Cardiology, Neuenheimer Feld 669, University Hospital, 69120 Heidelberg, Germany. Correspondence and requests for materials should be addressed to D.V. (email: dvilchez@uni-koeln.de)

Aprecisely coordinated network of transcriptional, chromatin and RNA modifiers regulate embryonic stem cell (ESC) identity. While the transcriptional, epigenetic, and signaling regulators of ESC function have been a primary focus of research efforts, emerging evidence indicates that post-transcriptional regulatory mechanisms also play a central role in ESC self-renewal, pluripotency and cell fate decisions[1–3]. However, the mechanisms by which post-transcriptional modulation impinges upon ESC identity and differentiation remain largely unknown. RNA-binding proteins (RBPs) participate in the regulation of practically any step of gene expression involving RNA, including transcription, alternative splicing, nuclear export, translation and turnover[2]. A study provided a census of 1542 manually curated RBPs in humans, ~7.5% of all protein-coding genes[4]. Using the "mRNA interactome capture" technique, which enables the identification of proteins bound to polyadenylated RNAs in vivo, 555 mRNA-binding proteins have been cataloged in mouse ESCs (mESCs)[5]. Although RBPs represent a significant percentage of all protein-coding genes in mammals, only a few of these RBPs such as LIN28, FOX2, and MBNL2 have been associated to ESC function and examined in detail in the context of pluripotency[2]. Recently, an analysis of 247 genes by using siRNAs has identified 16 novel RBPs involved in mESC pluripotency, including components of the small subunit processome that modulates 18S rRNA biogenesis[3].

LIN28 is one of the most studied RBPs in the context of pluripotency. Overexpression of either LIN28A or its paralog, LIN28B, enhances somatic reprogramming efficiency into induced pluripotent stem cells (iPSCs)[6]. LIN28 is highly expressed in pluripotent stem cells[7] to regulate the translation and stability of hundreds of mRNAs[8–10], modulating key biological processes such as metabolism[6]. LIN28 contains a cold shock domain (CSD), an ancient β-barrel that binds single-stranded nucleic acids[11]. CSD-containing proteins belong to the most evolutionarily conserved family of RBPs known among bacteria, plants, and animals[11]. In humans, the CSD-containing proteins include LIN28, the Y-box family (YBX1, YBX2, YBX3), a ribosomal RNA-processing protein (DIS3) and CSDE1. Although unrelated in their primary sequence, CSDs are structurally and functionally similar to S1 domains (also called "CSD-like" domain)[11]. In humans, the translation factors EIF1AX, EIF2A, EIF5A and the RNA-processing proteins DHX8, EXOSC3 and DIS3 contain S1 domains[11]. DHX8 has been identified as a potential regulator of ESCs in a RNAi screen against chromatin proteins[12]. Besides LIN28A, we have observed that other CSD and S1-containing proteins are highly expressed in hESCs. Thus, we performed a shRNA screen against these RBPs and found CSDE1 as an important determinant of hESC identity.

In contrast to other CSD-containing proteins, CSDE1 has multiple CSDs[11]. This RBP is mostly localized in the cytoplasm where it interacts with distinct complexes involved in the regulation of mRNA stability and/or translation[11]. Accordingly, CSDE1 post-transcriptionally regulates numerous mRNAs by different mechanisms depending on its interaction with other proteins, the target transcript and the region within the transcript where CSDE1 primarily binds in a dynamic process associated with the specific cell type and state. For instance, CSDE1 can either promote FOS mRNA turnover[13] or be part of a complex that stabilizes the parathyroid hormone (PTH) transcript[14]. Moreover, CSDE1 can modulate translation of its targets at different levels (i.e., initiation, elongation or termination) acting as either an activator or inhibitor of a specific translation phase for distinct mRNAs[15,16]. A paradigmatic case is the regulation of ribosome entry sites (IRES)-mediated translation, whereby sequence or structures located in the 5′UTR of specific mRNAs allow for translation initiation[17]. CSDE1 enhances IRES activity

of the pro-apoptotic factor Apaf-1[15] and the cell cycle PITSLRE kinase[18] stimulating their translation. On the contrary, binding to the IRES of its own transcript represses CSDE1 translation[17,19].

Given the versatile binding of CSDE1 to mRNA targets and other RBPs, CSDE1 coordinates multiple biological processes[11]. The complexity of this regulation underlies the ability of CSDE1 to modulate the same biological process in an opposing manner depending on the cell type and state[19–21]. CSDE1 can either promote or inhibit apoptosis[19] as well as differentiation in a cell-type specific manner. For instance, CSDE1 promotes erythroblast differentiation[21] whereas it prevents the differentiation of naive mESCs into extraembryonic primitive endoderm-like cells[20]. Another example of CSDE1 versatility has been extensively studied in Drosophila melanogaster. While CSDE1 represses X-chromosome dosage compensation in female Drosophila, it induces the opposite effect in males by sex-specific interactions with intrinsic target transcripts and RBPs[22–25]. In fact, the first systematic characterization of CSDE1 targets was performed in this organismal model by RNA immunoprecipitation (RIP) analysis[26]. These experiments revealed that CSDE1 binds to hundreds of transcripts such as mRNAs encoding developmental factors involved in TGF-beta (e.g., TGFB1) and WNT signaling pathways[26]. Recently, a comprehensive study combining individual-nucleotide resolution crosslinking immunoprecipitation sequencing (iCLIP-seq), RNA sequencing and ribosome profiling unveiled CSDE1 targets in human melanoma[16]. In these cells, CSDE1 protein expression is often increased and regulates the levels of pro-oncogenic factors such as vimentin (VIM) or RAC1 as well as tumor suppressors (e.g., PTEN)[16]. Interestingly, this study also demonstrated that CSDE1 binds to mRNAs encoding regulatory proteins involved in development and neuron projection guidance in melanoma cells[16].

Here we show that CSDE1 maintains the undifferentiated state of hESCs preventing neural differentiation, which is considered to be their default fate[27]. Accordingly, loss of CSDE1 accelerates neural differentiation and neurogenesis. Using a proteomics approach, we first identify that this process is partially modulated by post-transcriptional regulation of fatty acid binding protein 7 (FABP7) and VIM mRNAs. FABP7 and VIM are markers of radial glial cells, the neural progenitors that essentially generate, either directly or indirectly, most of the neurons in the mammalian brain[28]. FABP7 is required for brain development[29] and here we demonstrate that both FABP7 and VIM are essential for successful neurogenesis of hESCs. Moreover, we find that ectopic expression of CSDE1 decreases the levels of FABP7 and VIM, resulting in impaired neural differentiation. Concomitantly, CSDE1 modulates the transcript levels of core components of known regulatory nodes of hESC identity, neuroectoderm commitment and neuron differentiation. Taken together, our results establish CSDE1 as an essential post-transcriptional regulator of hESC fate decisions that can be modulated to promote neurogenesis.

## Results

**ESCs exhibit increased protein levels of CSDE1.** To examine the levels of CSD-containing proteins, we performed quantitative proteomics comparing hESCs with their differentiated neuronal counterparts. Besides LIN28A, we found that all the CSD and CSD-like proteins detected in our proteomics assay are significantly increased in hESCs (Supplementary Table 1 and Supplementary Data 1). Since LIN28A and DHX8 levels are linked to ESC function, we performed a shRNA screen against other CSD-containing proteins to identify potential novel regulators of hESC function. hESCs were infected with shRNA-expressing lentivirus and selected for puromycin resistance. Each knockdown (KD)

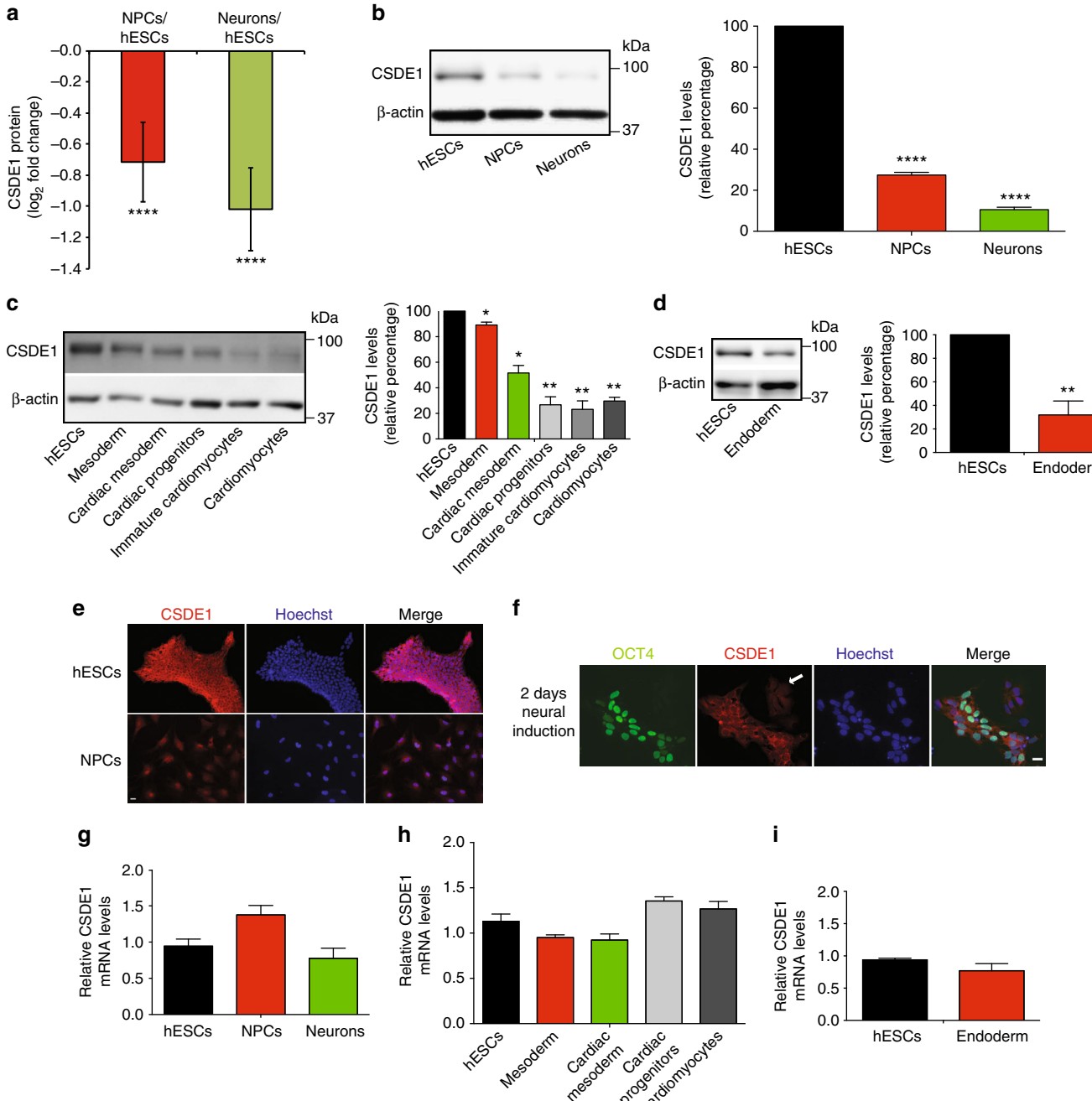

**Fig. 1** The levels of CSDE1 protein decrease during hESC differentiation. **a** Quantitative proteomic analysis of CSDE1 levels comparing H9 hESCs with their NPC and neuronal counterparts. Graph represents the mean (±confidence interval) of relative abundance differences calculated from the $\log_2$ of label-free quantification (LFQ) values (hESCs ($n = 9$), NPCs ($n = 6$) and neurons ($n = 5$)). Statistical comparisons were made by limma's moderated t-test (P-value: **** ($P < 0.0001$)). **b** Western blot analysis with antibody to CSDE1. The graph represents the CSDE1 relative percentage values (corrected for β-actin loading control) to H9 hESCs (mean ± s.e.m. of five independent experiments). **c** Western blot of CSDE1 in H1 hESCs and their differentiated mesoderm and cardiomyocyte counterparts. The graph represents the CSDE1 relative percentage values (corrected for β-actin loading control) to H1 hESCs (mean ± s.e.m. of two independent experiments). **d** Western blot analysis with antibody to CSDE1. The graph represents the CSDE1 relative percentage values (corrected for β-actin loading control) to H9 hESCs (mean ± s.e.m. of three independent experiments). **e** Immunocytochemistry of H9 hESCs and NPCs (2 weeks under neural induction treatment) with antibody to CSDE1. Hoechst staining was used as a marker of nuclei. Scale bar represents 20 μm. **f** Immunocytochemistry of cell cultures at early stages of the neural induction treatment (2 days) with antibody to CSDE1. OCT4 and Hoechst staining were used as markers of pluripotency and nuclei, respectively. These cultures contain undifferentiated (OCT4-positive) and differentiated (OCT4-negative) cells. White arrow indicates OCT4-negative cells. Scale bar represents 20 μm. **g** Quantitative PCR (qPCR) analysis of *CSDE1* mRNA levels. Graph (*CSDE1* relative expression to H9 hESCs) represents the mean ± s.e.m. of three independent experiments. **h** *CSDE1* relative expression to H1 hESCs represents the mean ± s.e.m. of three independent experiments with three biological replicates. **i** *CSDE1* relative expression to H9 hESCs represents the mean ± s.e.m. of two independent experiments with three biological replicates. In **b–d** and **g–i**, statistical comparisons were made by Student's t-test for unpaired samples. P-value: *($P < 0.05$), **($P < 0.01$), **** ($P < 0.0001$)

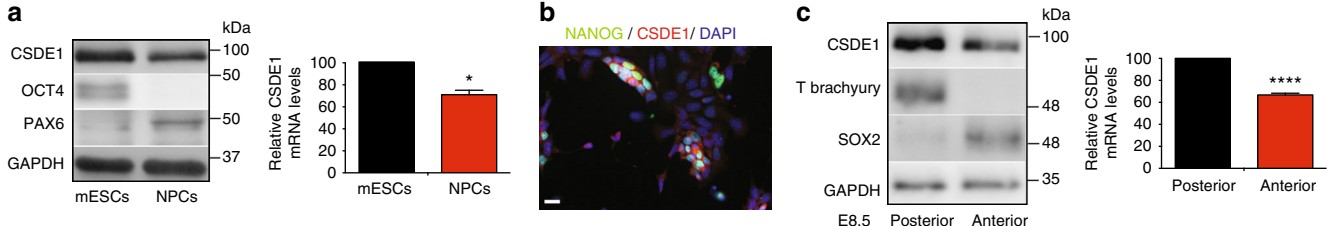

**Fig. 2** CSDE1 protein levels decrease with differentiation in developing mouse embryonic tissues. **a** Western blot analysis with antibody to CSDE1 of mouse ESCs (mESCs) and derived NPCs. The graph represents the CSDE1 relative percentage values (corrected for GAPDH loading control) to mESCs (mean ± s.e.m. of two independent experiments). **b** Immunocytochemistry of mESCs and differentiated cells with antibody to CSDE1. NANOG and DAPI staining were used as markers of pluripotency and nuclei, respectively. Scale bar represents 20 μm. **c** Western blot analysis of primitive streak and streak-enriched tissues (posterior) and neural plate and neural-enriched tissues (anterior) from mouse embryos (embryonic day (E)8.5)). T (brachyury) and SOX2 staining were used as markers of posterior and anterior parts, respectively. The graph represents the CSDE1 relative percentage values (corrected for GAPDH loading control) to posterior part (mean ± s.e.m. of three independent experiments). All the statistical comparisons were made by Student's t-test for unpaired samples. P-value: *(P < 0.05), **** (P < 0.0001)

hESC line was monitored daily (during 10 days) for alterations in cell or colony morphology. We did not observe significant differences in most of the KD hESCs (i.e., YBX1, YBX2, YBX3, DIS3, EIF1AX, EIF2A, EIF5A and EXOSC3) (Supplementary Fig. 1a). Accordingly, we did not find significant changes in the expression of pluripotency markers in these cells (Supplementary Fig. 1b). We only detected prominent morphological differences upon knockdown of CSDE1, indicating a potential role of this RBP in hESC function (Supplementary Fig. 2). Thus, we further assessed CSDE1 expression changes during differentiation. First, we examined CSDE1 protein levels using available quantitative proteomics data comparing hESCs with their differentiated neural progenitor cell (NPC) and neuronal counterparts[30] (Fig. 1a). Notably, hESCs lost their high CSDE1 levels when differentiated into NPCs (Fig. 1a) as we confirmed by western blot analysis (Fig. 1b and Supplementary Fig. 3). The downregulation in CSDE1 levels was not a specific phenomenon associated with the neural lineage as differentiation into other cell types also induced a decrease in CSDE1 protein amounts (Fig. 1c, d).

To assess the levels of CSDE1 in individual cells, we performed immunocytochemistry experiments. In hESCs, CSDE1 was mostly localized in the cytoplasm and, particularly, concentrated in the perinuclear region (Fig. 1e), as previously reported in other cell types[31]. When hESCs were differentiated into NPCs, the expression of CSDE1 was downregulated (Fig. 1e). As a more formal test, we examined cell populations at early stages of the neural induction treatment to have a heterogeneous culture with hESCs and differentiated cells. Notably, cells expressing pluripotency markers exhibited higher amounts of CSDE1 when compared to differentiated cells (Fig. 1f). Because CSDE1 is tightly regulated at the translational level[17,19], protein amounts may not correlate with mRNA levels[21]. Interestingly, we did not detect significant changes in CSDE1 mRNA levels during differentiation into the distinct cell types (Fig. 1g–i), indicating that downregulation of CSDE1 protein is modulated by post-transcriptional mechanisms.

With the strong connection between CSDE1 protein levels, pluripotency and differentiation, we asked whether the levels of CSDE1 changed during mouse neural development. After we confirmed that naive mESCs also have higher CSDE1 protein levels compared to their differentiated counterparts (Fig. 2a, b), we examined CSDE1 levels in developing mouse embryonic tissues. Remarkably, CSDE1 protein levels were lower in the developing neural plate and neural-enriched tissues compared with the primitive streak and streak-enriched tissues (Fig. 2c). Taken together, our results indicate that enhanced CSDE1 protein expression is associated with pluripotency and its levels decrease upon differentiation.

**CSDE1 prevents neural differentiation of hESCs**. Typical undifferentiated hESCs grow in tightly packed, three-dimensional colonies without spaces between cells. However, CSDE1 KD hESC cultures contained not only undifferentiated colonies but also monolayer colonies formed by flattened and elongated cells with reduced cell contact (Fig. 3a and Supplementary Fig. 2). This morphology contrasts with the growth pattern of control hESCs and other CSD-containing proteins KD hESCs, in which cells grew essentially in dense, three-dimensional colonies (Fig. 3a and Supplementary Figs. 1 and 2). Accordingly, CSDE1 KD cultures exhibited a decrease in alkaline phosphatase (AP) staining (Fig. 3b). Moreover, knockdown of CSDE1 also resulted in spontaneous neuronal differentiation (Fig. 3a). Given the strong phenotype observed in CSDE1 KD hESCs, we further characterized these cells. To maintain CSDE1 KD hESC lines, we transferred undifferentiated colonies followed by daily monitoring to remove differentiated cells (Supplementary Fig. 4). When we grew hESCs without removing differentiated colonies, flattened cells proliferated in CSDE1 KD lines resulting in decreased levels of pluripotency markers such as OCT4, NANOG and DPPA2 compared with control hESCs (Fig. 3c, d). In addition, we observed a significant decrease in the percentage of OCT4-positive cells in CSDE1 KD lines (Fig. 3e, f). Although loss of CSDE1 had no effect on hESC proliferation (Supplementary Fig. 5) as previously observed in mESCs[32], our results revealed profound differences between these cells regarding cell fate decisions triggered by CSDE1 downregulation. mESCs are in a more naive state than hESCs and retain their ability to differentiate into primitive endoderm[33], an extraembryonic tissue that maintains the expression of pluripotency markers while inducing high levels of endodermal markers such as GATA6[20,34]. Whereas it has been reported that loss of CSDE1 induces the proliferation of primitive endoderm in mESC cultures[20], we found that CSDE1 downregulation in hESCs results in differentiated cells that lose the expression of pluripotency markers (Fig. 3c–f). Moreover, we did not observe an induction of GATA6 and other endodermal markers in hESCs (Supplementary Fig. 6).

Although loss of CSDE1 induced differences in cell morphology as well as decreased expression of distinct core pluripotency markers, the levels of SOX2 were not altered in CSDE1 KD cells (Fig. 3c, d). SOX2 is not only highly expressed in hESCs but also in neural stem and progenitor cells[35]. As we observed a spontaneous differentiation into neuronal cells of CSDE1 KD cultures (Fig. 3a), we hypothesized that loss of CSDE1 commits hESCs to a neuroectoderm fate. In support of this hypothesis, we found that CSDE1 downregulation induces the proliferation of cells expressing PAX6, an early marker of neuroectodermal differentiation (Fig. 3e, f). We tested three independent shRNAs

to CSDE1 and obtained similar results (Fig. 3e, f and Supplementary Fig. 7). Likewise, knockout of CSDE1 (Supplementary Fig. 8a) also resulted in the proliferation of PAX6-positive cells (Fig. 3g). Because hESC lines can vary in their characteristics, we examined whether loss of CSDE1 induces proliferation of PAX6-positive cells in distinct lines. Indeed, knockdown of CSDE1 (Supplementary Fig. 8b–d) triggered spontaneous differentiation into OCT4-lacking cells that express

high levels of PAX6 in all the lines tested (Fig. 3h–j). Overall, these results indicate that the intrinsic high levels of CSDE1 in hESCs prevent their neural differentiation.

**Loss of CSDE1 potentiates neural differentiation.** With the spontaneous differentiation of CSDE1 KD hESCs into PAX6-positive cells, we asked whether this RBP is involved in the

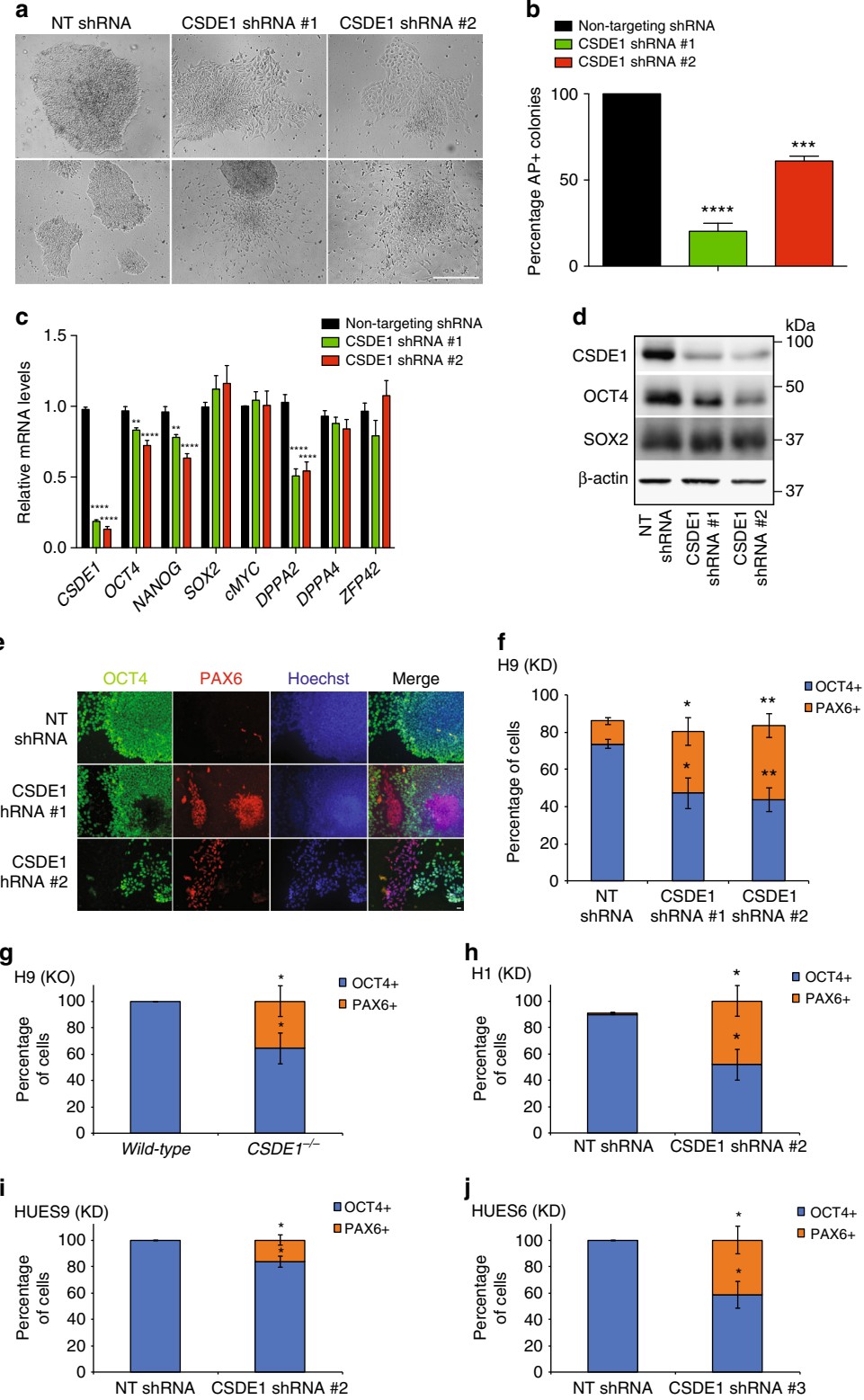

regulation of neural differentiation. To test this hypothesis, we selected undifferentiated hESC colonies and induced differentiation into NPCs (Fig. 4a). We found that CSDE1 KD hESCs differentiate significantly faster into PAX6-positive cells than control hESCs (Fig. 4a and Supplementary Fig. 9). We obtained similar results with *CSDE1*[−/−] hESCs (Supplementary Fig. 10). Likewise, other independent hESC lines as well as iPSCs also showed a faster neural differentiation on CSDE1 knockdown (Supplementary Fig. 11). At the end of the neural induction treatment, both control and CSDE1 KD cultures consisted mostly of PAX6-positive cells (Fig. 4a). However, CSDE1 KD NPCs exhibited higher expression of PAX6 and other neural markers at this stage (Fig. 4b and Supplementary Fig. 12). One step further was to determine whether these cells are able to generate terminally differentiated neurons. For this purpose, we induced neuronal differentiation of CSDE1 KD NPCs. After the first 10 days of neuronal induction, CSDE1 KD cells expressed higher levels of neuronal markers (e.g., neurofilaments, synaptic proteins) than control cells (Fig. 4c). Consistent with this enhanced induction of neuronal markers, we observed increased neurogenesis in CSDE1 KD lines during the first week of differentiation (Fig. 4d). After three weeks of neuronal induction, CSDE1 KD cells were differentiated almost exclusively into MAP2-positive cells (Fig. 4d). In contrast, control NPCs had decreased efficiency in neurogenesis and generated a significant percentage of astrocytes when compared to CSDE1 KD cells (Fig. 4d).

Since our results indicated that loss of CSDE1 in hESCs facilitates their neural differentiation, we assessed the potential of CSDE1 KD hESCs to differentiate into other cell types. Notably, knockdown of CSDE1 reduced the induction of endoderm markers upon definitive endodermal differentiation when compared with control cells (Fig. 4e). Similarly, CSDE1 KD hESCs exhibited a diminished induction of cardiac mesoderm and cardiomyocytes markers during their progressive differentiation into cardiomyocytes (Supplementary Fig. 13). Taken together, our data suggest that loss of CSDE1 commits hESCs to neuroectoderm cell fate.

**CSDE1 post-transcriptionally regulates *FABP7* and *VIM*.** CSDE1 binds to complexes involved in the regulation of mRNA stability and translation[11]. To determine the protein binding partners of CSDE1 in hESCs, we performed co-immunoprecipitation experiments (Supplementary Fig. 14a) followed by a single shot label-free proteomic approach (Supplementary Table 2 and Supplementary Data 2). Notably, we found a novel interaction of CSDE1 with the 40S ribosomal protein S27 (RPS27) (Supplementary Table 2). Accordingly, CSDE1 protein is markedly present in 40S fractions as assessed by ribosome fractionation experiments (Supplementary Fig. 14b). Moreover, we found that CSDE1 co-immunoprecipitated with Pumilio homolog 1 (PUM1) (Supplementary Table 2), a RBP that binds the 3′UTR of specific mRNA targets mediating their post-transcriptional repression through translational inhibition and mRNA degradation[36]. In addition, Gene Ontology (GO) analysis of CSDE1 potential interactors indicated enrichment for proteins associated with mRNA cap binding complex (Supplementary Fig. 14c). Since we did not observe differences in global translation rates upon knockdown of CSDE1 in hESCs (Fig. 5a), we hypothesized that this RBP post-transcriptionally regulates specific mRNAs involved in neuroectodermal commitment and neurogenesis.

To examine the mechanism(s) by which CSDE1 regulates neural differentiation, we analyzed undifferentiated CSDE1 KD hESC colonies. Hereby, we avoided the use of heterogeneous populations containing cells in advance states of differentiation that could mask the core early events by which loss of CSDE1 triggers neural differentiation. First, we confirmed by western blot that these cells express similar levels of OCT4 and PAX6 compared with control hESCs (Fig. 5b). Furthermore, we did not observe increased levels of markers of the distinct germ layers (Supplementary Fig. 15). Then, we performed quantitative proteomics to analyze their proteome. Besides CSDE1 levels, quantitative proteomics analysis revealed that other 20 proteins (out of 4435 quantified proteins in all the samples) are significantly changed in both independent CSDE1 shRNA hESC lines (Supplementary Table 3 and Supplementary Data 3). Among them, 10 proteins were up-regulated (e.g., FABP7 and VIM) whereas the others were down-regulated (e.g., THUMPD3). The transcripts of 6 significantly changed proteins (VIM, DNM1, SMARCC2, CSDE1, EIF4A2 and ANXA2) have been previously identified as direct RNA targets of CSDE1 in human melanoma cells[16]. In the context of neural differentiation, it is particularly relevant that CSDE1 induced upregulation of VIM (Supplementary Table 3), an intermediate filament subunit highly expressed in neuroepithelial and radial glial cells[37,38]. Interestingly, the most up-regulated protein in CSDE1 KD hESCs was FABP7 (also known as brain lipid-binding protein (BLBP)), a fatty acid binding protein required for the maintenance of neuroepithelial cells and neurogenesis from radial glia during brain development[29,38,39]. Similar to radial glial cells, we found that FABP7 is also highly expressed in NPCs derived from control hESCs (Supplementary Fig. 16). We identified five consensus binding motifs of CSDE1 in the *FABP7* transcript (Supplementary Data 4), supporting a potential role of this RBP in FABP7 post-transcriptional regulation.

Western blot experiments confirmed increased protein levels of FABP7 upon knockdown of CSDE1 in four independent hESC lines as well as iPSCs, even when we did not observe changes in the early neural marker PAX6 (Fig. 5b and Supplementary Fig. 17). Furthermore, the mRNA levels of *FABP7* were increased in CSDE1 KD hESCs (Fig. 5c). Similarly, increased VIM protein amounts were correlated with

**Fig. 3** CSDE1 prevents neural differentiation of hESCs. **a** Brightfield images of H9 hESCs. Knockdown of CSDE1 results in the proliferation of flattened and elongated cells that grew in monolayer colonies with reduced cell contact. Furthermore, loss of CSDE1 induces a spontaneous differentiation into neuronal cells. Scale bar represents 250 μm. **b** Percentage of alkaline phosphatase (AP)-positive colonies after five days of culturing without removing differentiated cells. Graph represents the mean ± s.e.m. of the percentage observed in four independent experiments (we assessed approximately 150 total colonies in each independent experiment). **c** qPCR analysis of hESCs cultured for five days without removal of differentiated cells. Graph (relative expression to non-targeting (NT) shRNA) represents the mean ± s.e.m. of four independent experiments with three biological replicates. **d** Western blot analysis with antibodies to CSDE1, OCT4 and SOX2. β-actin is the loading control. hESCs were grown for five days without removing differentiated cells. **e** Immunocytochemistry of H9 hESCs grown for five days without removal of differentiated cells. OCT4, PAX6, and Hoechst staining were used as markers of pluripotency, neuroectodermal differentiation, and nuclei, respectively. Scale bar represents 20 μm. **f–j** Graphs represent the percentage (mean ± s.e.m.) of OCT4 and PAX6-positive cells/total nuclei after five days in culture without removal of differentiated cells: **f** H9 hESCs, n = 3 independent experiments, 550–700 total cells per experiment; **g** H9 hESCs, n = 3, 420–1000 cells per experiment; **h** H1 hESCs, n = 3, 290–960 cells per experiment; **i** HUES9 hESCs, n = 3, 320–710 cells per experiment; **j** HUES6 hESCs, n = 3, 200–550 cells per experiment. KD = *CSDE1* knockdown. KO = *CSDE1* knockout. All the statistical comparisons were made by Student's t-test for unpaired samples. P-value: *(P < 0.05), **(P < 0.01), ***(P < 0.001), **** (P < 0.0001)

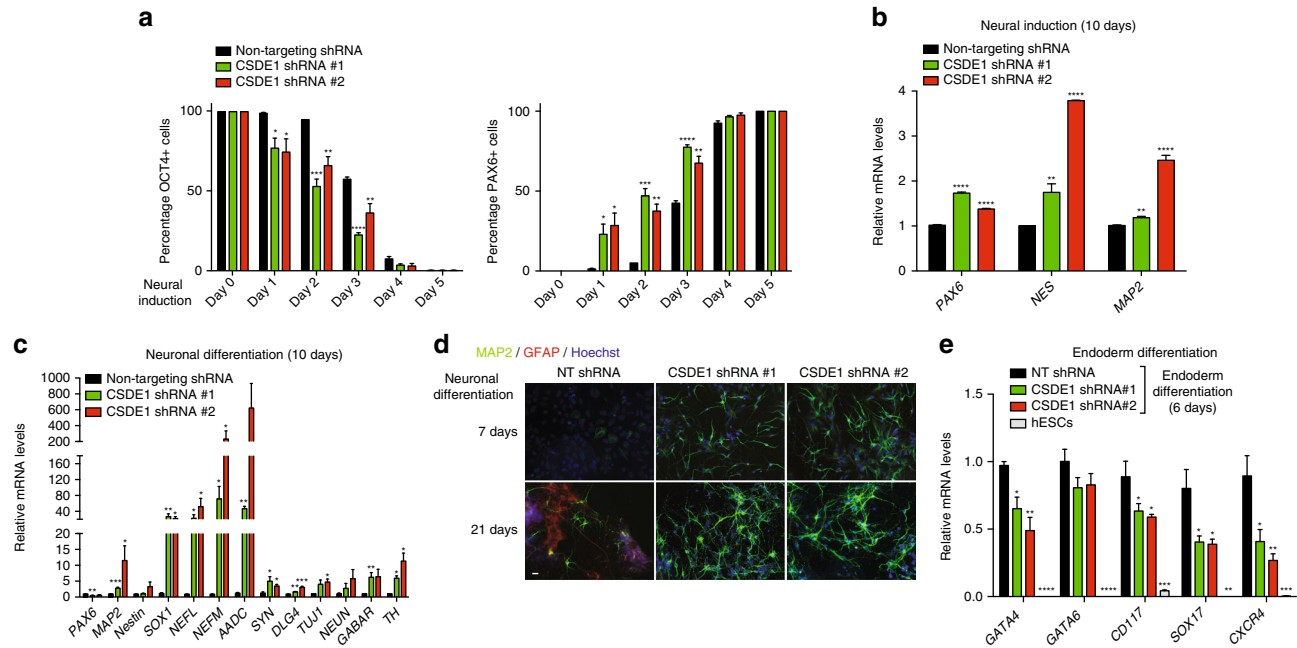

**Fig. 4** Loss of CSDE1 promotes neural differentiation and neurogenesis. **a** Percentage of OCT4 and PAX6-positive cells/total nuclei at different days after neural induction of undifferentiated H9 hESC colonies (mean ± s.e.m. of three independent experiments, 200 total cells per experiment). **b** qPCR analysis after 10 days of neural induction. Data (relative expression to NT shRNA cells) represent two independent experiments with three biological replicates. **c** qPCR analysis of neuronal markers after 10 days of pan-neuronal differentiation. Graph (relative expression to NT shRNA) represents the mean ± s.e.m. of three independent experiments with three biological replicates. **d** Immunocytochemistry after 7 and 21 days of neuronal induction. MAP2, GFAP, and Hoechst staining were used as markers of neurons, astrocytes, and nuclei, respectively. Scale bar represents 20 μm. **e** qPCR analysis of endoderm markers upon definitive endodermal differentiation of undifferentiated H9 hESCs. Graph (relative expression to NT shRNA) represents the mean ± s.e.m. of three independent experiments with two biological replicates. All the statistical comparisons were made by Student's t-test for unpaired samples. P-value: *(P < 0.05), **(P < 0.01), ***(P < 0.001), **** (P < 0.0001)

upregulation of mRNA levels (Fig. 5b, c). Taken together, our data links CSDE1 levels with modulation of FABP7 and VIM, proteins highly expressed in NPCs, neuroepithelial and radial glial cells. Radial glia not only exhibit properties of their precursor neuroepithelial cells but also astroglial characteristics[38]. Although CSDE1 KD hESCs had enhanced levels of FABP7, they did not exhibit significant changes in astroglial markers that are normally expressed in radial glia such as glial fibrillary acidic protein (GFAP) or astrocyte-specific glutamate transporter (SLC1A3) (Supplementary Fig. 18). Thus, these results suggest that knockdown of CSDE1 increases the levels of FABP7 and VIM in hESCs without triggering global changes in astrocyte markers.

With the strong connection between the levels of CSDE1 and FABP7, an essential protein for brain development[29], we examined whether CSDE1 protein binds to FABP7 mRNA by performing RIP assays. Given that CSDE1 protein directly interacts with its own mRNA in vivo to regulate its stability and translation[19], we assessed CSDE1 mRNA as a positive control (Fig. 5d). Besides CSDE1 mRNA, RIP experiments indicated that CSDE1 protein also binds to FABP7 and VIM mRNA in hESCs (Fig. 5d). Moreover, we found that in vitro transcribed biotinylated VIM and FABP7 mRNAs pull down CSDE1 protein from hESC lysates (Supplementary Fig. 19). To determine whether CSDE1 modulates the stability of these mRNAs, we performed actinomycin D chase experiments. We found that loss of CSDE1 decreases the degradation of both FABP7 and VIM mRNAs (Fig. 5e). On the contrary, knockdown of CSDE1 induced a slight increase in the degradation of NANOG mRNA whereas it did not affect the stability of other mRNAs such as OCT4, SOX2, PAX6, DPPA2, or THUMPD3 (Fig. 5e). Besides mRNA stability, we assessed whether CSDE1 regulates the

translation of FABP7 and VIM by ribosome fractioning experiments followed by quantitative PCR. Whereas the levels of VIM and other mRNAs (e.g., THUMPD3, PAX6, OCT4) in polysome fractions correlated with those observed in total cell extracts (Fig. 5f), the amount of FABP7 mRNA was further increased in polysome fractions of CSDE1 KD hESCs (Fig. 5f). Overall, our data indicate that CSDE1 regulates both the stability and translation of FABP7 mRNA as well as the stability of VIM mRNA in hESCs.

**Ectopic expression of CSDE1 impairs neural differentiation.** The neural fate is considered to be the intrinsic commitment of ESC differentiation. Given that our results indicate that CSDE1 prevents neural differentiation of hESCs, we asked whether post-transcriptional regulation of FABP7 and VIM contributes to this process. Fabp7$^{-/-}$ mice have decreased number of neural stem cells and neurogenesis in the developing brain[40]. In hESCs, knockdown of FABP7 did not change the levels of pluripotency markers (Fig. 6a). Upon neural differentiation induction, FABP7 KD hESCs differentiated significantly slower into PAX6-positive cells than control hESCs (Fig. 6b). At the end of the neural differentiation treatment, FABP7 KD cultures exhibited a mild decreased induction of PAX6-positive cells (Fig. 6c, d), as well as diminished up-regulation of neural (NESTIN) and neuronal (MAP2) markers (Fig. 6e). Moreover, loss of FABP7 blocked the capacity of these cells to generate terminally differentiated neurons while maintaining their ability to differentiate into astrocytes (Fig. 6f, g). Similarly, loss of VIM also resulted in impairment of neuronal differentiation (Fig. 6f, g). Taken together, these results suggest that FABP7 and VIM modulate neural differentiation of hESCs.

Since the levels of CSDE1 decreased during hESC differentiation (Fig. 1), we asked whether ectopic expression of this RBP prevents neural differentiation. We found that mild overexpression of CSDE1 further decreased the transcript and protein levels of FABP7 and VIM in hESCs (Fig. 7a, b). Upon neural induction, overexpression of CSDE1 resulted in diminished generation of PAX6-positive cells (Fig. 7c–f). Collectively, our data indicate CSDE1 as a negative regulator of FABP7 and VIM levels in hESCs, a mechanism that could contribute to prevent their neuroectoderm fate.

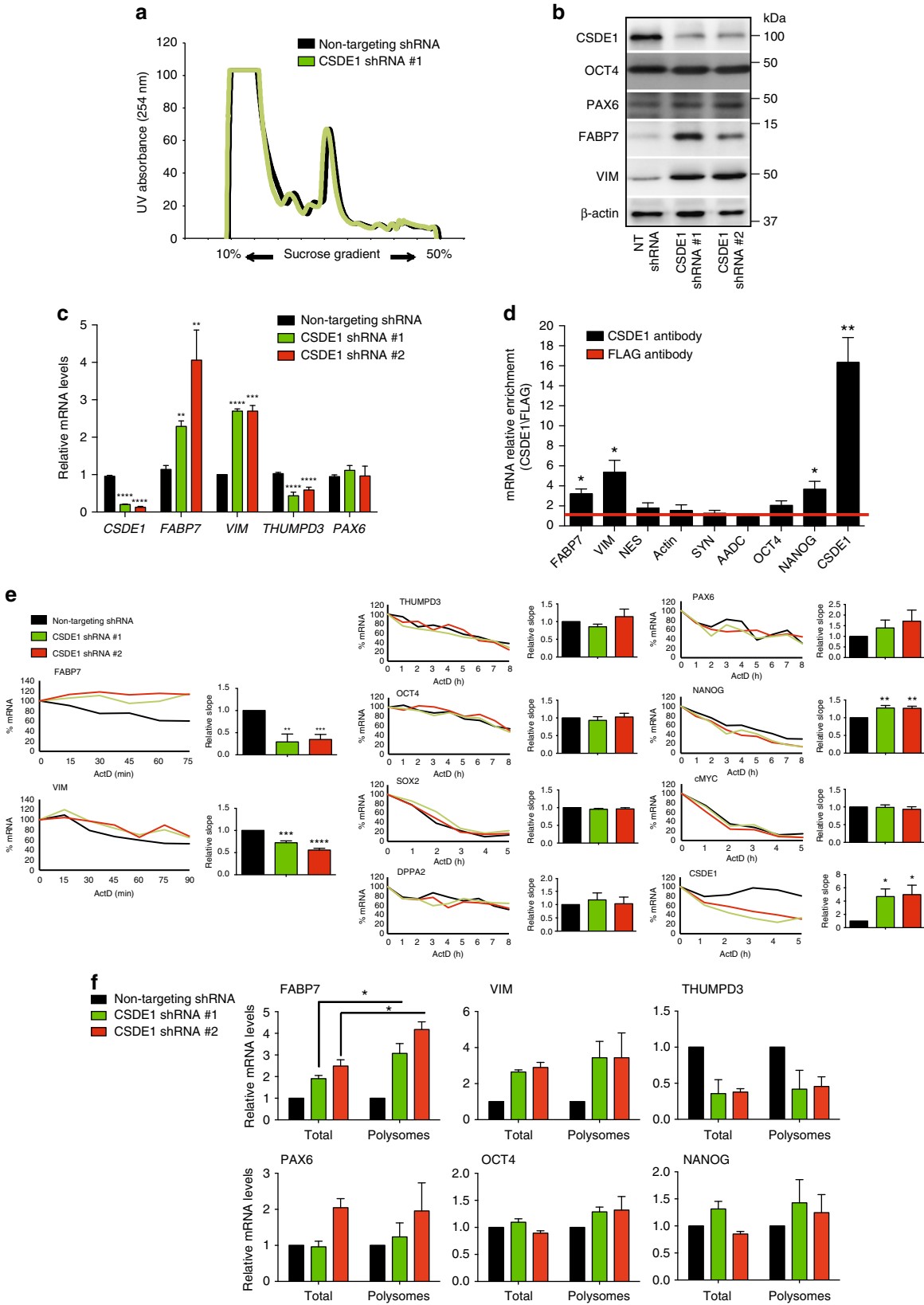

**CSDE1 prevents pro-neural changes in the hESC transcriptome**. CSDE1 can regulate multiple signaling pathways through its interaction with hundreds of different transcripts[11]. Besides FABP7 and VIM, we hypothesized that additional mechanisms could contribute to neural differentiation on CSDE1 knockdown. Whereas proteomic analysis is a valuable approach to identify large changes in protein levels regulated by CSDE1 (Supplementary Data 3), it also presents important limitations for our study. For instance, this approach might restrict the quantification of regulatory components such as transcriptions factors as well as low abundant proteins in hESCs which expression is triggered during differentiation. Thus, we performed a transcriptomic analysis of undifferentiated colonies to further identify changes in regulators of neural differentiation induced by CSDE1 downregulation. Since CSDE1 binds primarily mature mRNAs[16], we focused on poly(A) transcripts. Besides *FABP7* and *VIM*, transcriptome analysis revealed that the steady-state levels of other 1452 transcripts (out of 24885 identified transcripts) are significantly changed in both independent CSDE1 shRNA hESC lines (Supplementary Data 5). Among them, we identified 1207 transcripts with at least one consensus CSDE1 binding motif (Fig. 8a and Supplementary Data 4) of which 531 transcripts were up-regulated whereas 676 were down-regulated (Fig. 8b). GO biological process (GOBP) term analysis indicated strong enrichment for genes involved in the regulation of organismal development, cell differentiation, neurogenesis (e.g., *DDIT4, EPHA2, TRPC6, FZD2, FZD7*) and neuron projection development/guidance (e.g., *NEFM, DAB2IP, EPHB3, SEMA4A, SEMA6C, EFNB1*) (Fig. 8c, d and Supplementary Data 6). Moreover, we observed GOBP enrichment for modulators of the WNT signaling pathway (*e.g., APCDD1, TMEM64, FZD7, GPC4, CDH2*), a regulatory node of pluripotency and neural differentiation[41,42] (Fig. 8c, d and Supplementary Data 6). Besides WNT signaling regulators, we also found changes in core components of known regulatory nodes of hESC identity and neural differentiation[41] such as the BMP signaling pathway (e.g., *BMP4, BMPR1A, ROR2*), the FGF receptor signaling pathway (i.e., *FGFR2, FGFR3, FGFR4*), the TGF-beta/SMAD binding signaling pathway (e.g., *FOXH1* and the activin A receptor *ACVR2B*) and *LIN28A*. Moreover, we observed changes in the steady-state transcript levels of genes involved in insulin/insulin-like growth factor binding (e.g., *INSR, IGFBP5*). Despite these alterations in numerous transcript levels, we did not observe downregulation of the pluripotency factor *NANOG* in our transcriptome analysis (Supplementary Data 5).

We selected over 90 known regulators of neurogenesis and signaling pathways involved in the neural commitment of hESCs for further validation by qPCR (Supplementary Data 7). We confirmed significant changes in approximately a 50% out of the selected transcripts (Supplementary Data 7). Moreover, the steady-state levels of these transcripts were also impaired in other hESC lines as well as iPSCs upon CSDE1 knockdown (Fig. 9a–d and Supplementary Data 7). Among the transcripts significantly changed in three or more independent pluripotent stem cells we found a downregulation of *DDIT4*, an inhibitor of neuronal differentiation and neurite outgrowth[43] previously identified as a binding target of CSDE1 in melanoma[16]. Moreover, we observed consistent changes in regulatory factors involved in the development of neuron projections such as increased *EPHB3* mRNA levels, an ephrin type-B receptor that functions in axon guidance and promotes development/maturation of dendritic spines[44,45]. In all the pluripotent stem cell lines tested, loss of CSDE1 induced downregulation of *SEMA4A*, an inhibitor of axonal extension[46,47] (Fig. 9a–d and Supplementary Data 7). In addition, the levels of specific WNT signaling regulators (i.e., *CDH2, TMEM64, GPC4*), *BMP4*, the insulin receptor *IGFBP5* as well as TGF-beta pathway modulators (i.e., *FOXH1, ACVR2B*) were changed in at least three pluripotent stem cell lines on CSDE1 knockdown (Fig. 9a–d and Supplementary Data 7). We confirmed that *CSDE1*$^{-/-}$ hESCs also exhibit alteration in the steady-state levels of these transcripts whereas the expression of the pluripotency marker *OCT4* was similar compared to control hESCs (Fig. 9e and Supplementary Data 7). Consistently, mild overexpression of CSDE1 resulted in significant changes in the levels of specific transcripts such as *SEMA4A, CDH2, TMEM64* and *BMP4* (Fig. 9f and Supplementary Data 7). To determine whether these transcripts are direct CSDE1 targets, we performed RIP experiments. As a positive control, we examined *DDIT4, CDH2* and *TMEM64* mRNAs that have been previously reported to interact with CSDE1 protein in melanoma[16]. Besides these transcripts, we found that CSDE1 also binds *SEMA4A, GPC4, BMP4, FOXH1, EPHB3* and *APCDD1* mRNAs in hESCs (Fig. 10a). In contrast, we did not observe interaction of CSDE1 protein with *FZD7, SEMA6C, ACVR2B* or *IGFBP5*, suggesting significant secondary effects induced by CSDE1 knockdown (Fig. 10a), consistent with previous reports[16]. Among all the transcripts tested, *SEMA4A* was the most enriched mRNA upon CSDE1 immunoprecipitation (Fig. 10a). Since decreased *SEMA4A* mRNA levels resulted in downregulation of its protein expression (Fig. 10b and Supplementary Fig. 20), we examined whether CSDE1 post-transcriptionally regulates *SEMA4A* as a further validation of CSDE1 function in neural differentiation of hESCs. We found that loss of CSDE1 decreases *SEMA4A* mRNA stability (Fig. 10c). The mRNA levels of *SEMA4A* in polysome fractions correlated with those observed in total cell extracts, indicating that post-transcriptional regulation of this transcript by CSDE1 is performed at the mRNA turnover level (Fig. 10d). Moreover, we confirmed that other CSDE1 potential targets (i.e., *EPHB3, CDH2*) are also modulated at the mRNA stability level in hESCs (Fig. 10c, d). Taken together,

**Fig. 5** Knockdown of CSDE1 impairs post-transcriptional regulation of FABP7 and VIM. **a** Polysome profiles indicate no differences in the ribosome pool upon CSDE1 knockdown (graph is representative of 3 independent experiments). **b** Western blot analysis with antibodies to CSDE1, OCT4, PAX6, FABP7 and VIM of H9 hESCs daily monitored to remove differentiated cells. β-actin is the loading control. **c** Graph (relative expression to NT shRNA H9 hESCs) represents the mean ± s.e.m. of three independent experiments with three biological replicates. **d** Ribonucleoprotein immunoprecipitation (RIP) with CSDE1 antibody. Quantitative PCR analysis of the indicated genes is expressed as fold enrichment over RIP performed with FLAG control antibody. Graph (relative enrichment to FLAG antibody) represents the mean ± s.e.m. ($n = 4$ independent experiments). **e** mRNA levels were determined after the indicated time of actinomycin D (ActD) treatment (5 μg ml$^{-1}$) by qPCR. For each gene, the right graph corresponds to a representative experiment showing the percentage of mRNA relative to time = 0. In the left panel, mRNA degradation is shown as relative slope to non-targeting (NT) shRNA hESCs (mean ± s.e.m. of five independent experiments). The time course experiments with ActD treatment were established for every gene depending on the mRNA degradation rates observed in the NT shRNA samples. **f** Polysome profiling experiments followed by qPCR analysis. Total and polysome fractions mRNA levels are expressed as relative values to total and polysome fractions of NT shRNA hESCs, respectively. Graph represents the mean ± s.e.m. ($n = 4$ independent experiments). In **c**, **d** and **e** the statistical comparisons were made by Student's t-test for unpaired samples. In **f** the statistical comparisons were made by Student's *t*-test for paired samples (the mRNA polysome fraction was paired to its corresponding total mRNA in each independent experiment). *P*-value: *($P < 0.05$), **($P < 0.01$), ***($P < 0.001$), **** ($P < 0.0001$)

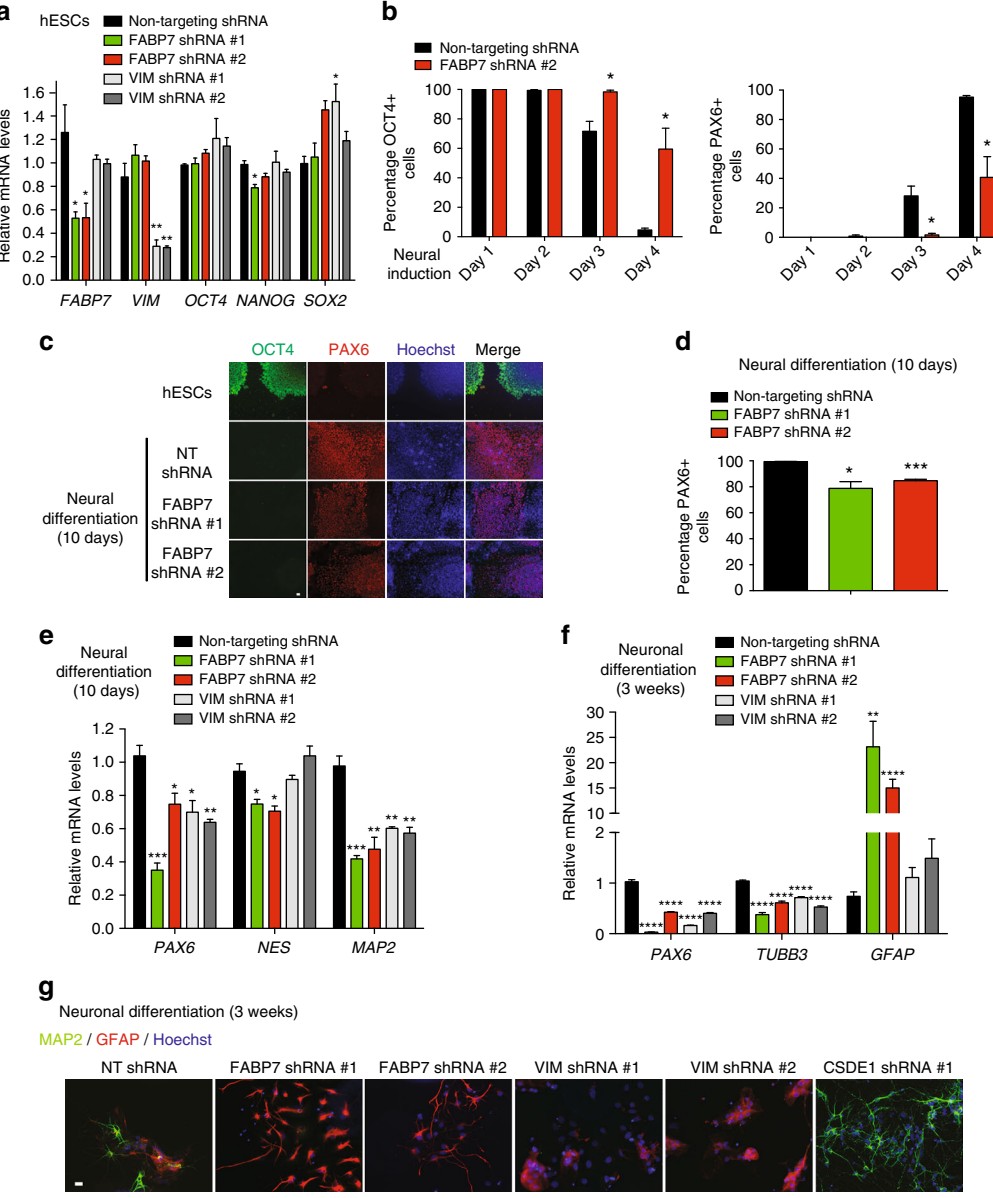

**Fig. 6** Loss of FABP7 in hESCs reduces their neural differentiation potential. **a** qPCR analysis in FABP7 and VIM KD H9 hESCs. Graph (relative expression to NT shRNA H9 hESCs) represents the mean ± s.e.m. of two independent experiments with three biological replicates. **b** Knockdown of FABP7 slows down neural differentiation. Percentage of OCT4 and PAX6-positive cells/total nuclei at different days after neural induction of H9 hESCs (mean ± s.e.m. (n = 3, 400–650 total cells per data point). **c** After 10 days of neural induction, cells were assessed by immunofluorescence with OCT4, PAX6, and Hoechst staining. Scale bar represents 20 μm. **d** Quantification of the percentage of PAX6-positive cells/total nuclei after 10 days of neural induction. Graph represents the mean ± s.e.m. of 3 independent experiments, 3000 total cells per experiment. **e** qPCR analysis after 10 days of neural induction. Graph (relative expression to NT shRNA cells) represents the mean ± s.e.m. of three independent experiments. **f** qPCR analysis after 3 weeks of neuronal induction. Data (relative expression to NT shRNA cells) represent the mean ± s.e.m. of three independent experiments with three biological replicates. **g** After 3 weeks of neuronal induction, cells were assessed by immunofluorescence with MAP2, GFAP, and Hoechst staining. Scale bar represents 20 μm. All the statistical comparisons were made by Student's t-test for unpaired samples. P-value: *(P < 0.05), **(P < 0.01), ***(P < 0.001), **** (P < 0.0001)

our results indicate that loss of CSDE1 changes the transcriptome landscape of hESCs making these cells more prone to neural differentiation.

## Discussion

Pluripotent stem cells hold a great promise for regenerative medicine. Moreover, these cells represent an invaluable resource to investigate human diseases and development. Thus, defining the regulatory mechanisms of pluripotency and transitions to differentiated cells is of central importance. Given the key role of

LIN28 in pluripotency, we conducted a shRNA screen to determine whether other CSD-containing RBPs regulate hESC function. Our results identified CSDE1 as a determinant of hESC identity. Although we did not observe changes in cell morphology or pluripotency markers upon knockdown of other CSD-containing proteins, we cannot discard that these RBPs also play an important role in cell fate decisions. Thus, further analysis will be required to assess a potential role of the distinct CSD-containing proteins in differentiation.

Given the strong phenotype observed upon CSDE1 knockdown, we focused on understanding the molecular mechanisms

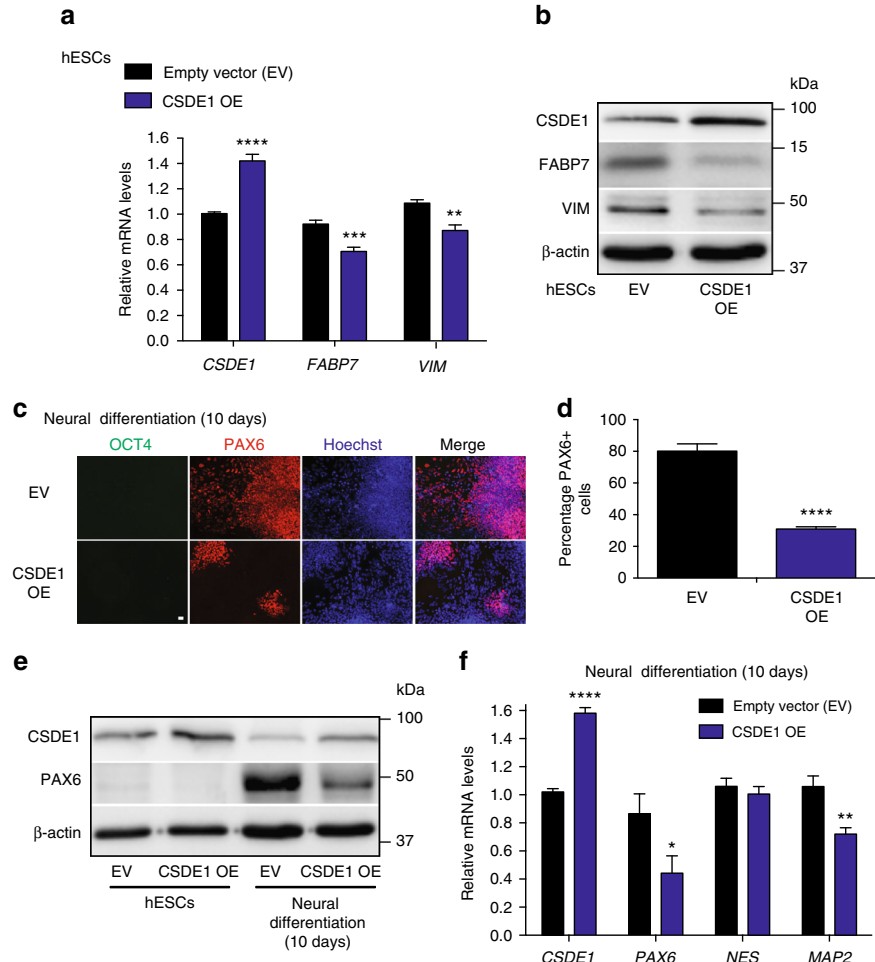

**Fig. 7** Ectopic expression of CSDE1 reduces FABP7 levels and impairs neural differentiation. **a** qPCR analysis of FABP7 and VIM levels (relative expression to EV H9 hESCs) in CSDE1-overexpressing (CSDE1 OE) H9 hESCs. Data represent two independent experiments with three biological replicates. **b** Western blot analysis of H9 hESCs with antibodies to CSDE1, FABP7 and VIM. β-actin is the loading control. **c** After 10 days of neural induction, cells were assessed by immunofluorescence with OCT4, PAX6, and Hoechst staining. Scale bar represents 20 μm. **d** Quantification of the percentage of PAX6-positive cells/total nuclei after 10 days of neural induction. Graph represents the mean ± s.e.m. of three independent experiments, 1000–1400 total cells per experiment. **e** Western blot analysis of H9 hESCs and their differentiated counterparts after 10 days of neural induction. **f** qPCR analysis after 10 days of neural induction. Data (relative expression to EV shRNA cells) represent three independent experiments with three biological replicates. All the statistical comparisons were made by Student's t-test for unpaired samples. P-value: *(P < 0.05), **(P < 0.01), ***(P < 0.001), ****(P < 0.0001)

by which CSDE1 regulates hESC function. Interestingly, CSDE1 protein is highly abundant in hESCs and its levels decrease with differentiation into the distinct germ layers. However, the decrease in CSDE1 protein levels does not correlate with changes in mRNA expression. Thus, our results indicate that down-regulation of CSDE1 protein expression during differentiation is regulated via translational or post-translational mechanisms. As other CSD-containing proteins, CSDE1 is mainly regulated at the post-transcriptional level[19]. CSDE1 negatively regulates its own translation by binding to an IRES in the 5′ UTR of CSDE1 transcript. Besides CSDE1 itself, other proteins interact with the IRES of the CSDE1 transcript to either repress or enhance translation[17,19]. For instance, polyprimidine tract-binding protein (PTB) inhibits CSDE1 translation whereas hnRNP C1/C2 (HNRNPC) and specific ribosomal subunits stimulate CSDE1 protein expression[17,21,48]. Since IRES regulators bind to CSDE1 transcript in a dynamic process[17], it will be fascinating to define modulators of CSDE1 levels during differentiation.

Notably, increased CSDE1 levels maintain hESC function by preventing neural differentiation. Since the neuroectoderm fate is considered to be the default commitment of ESCs[27], high levels of

CSDE1 could contribute to halt their intrinsic neural fate and maintain pluripotency. Our results are consistent with the essential role of CSDE1 in development as CSDE1-deficient mouse embryos die around mid-gestation[49]. In addition, we found that loss of CSDE1 accelerates neural differentiation and promotes neurogenesis. Remarkably, CSDE1 KD NPCs differentiate almost exclusively in neurons and exhibit decreased proliferation of astrocytes. Thus, our results suggest that CSDE1 modulation is a valuable approach for stem cell therapies in regenerative medicine. Whereas our findings establish an inverse correlation between CSDE1 levels and neural differentiation, CSDE1 may also modulate other cell fate decisions or characteristics of ESCs and germ-layer cells. In this regard, we have observed that CSDE1 binds PUM1, a RBP involved in ESC self-renewal that accelerates the downregulation of distinct mRNAs encoding pluripotency transcription factors[50]. In addition, knockdown of CSDE1 negatively affects hESC differentiation into distinct cell types such as definitive endoderm and cardiomyocytes. A potential role of CSDE1 in mesoderm development is supported by the phenotype reported in Csde1[-/-] mouse embryos, which exhibit delayed heart maturation with

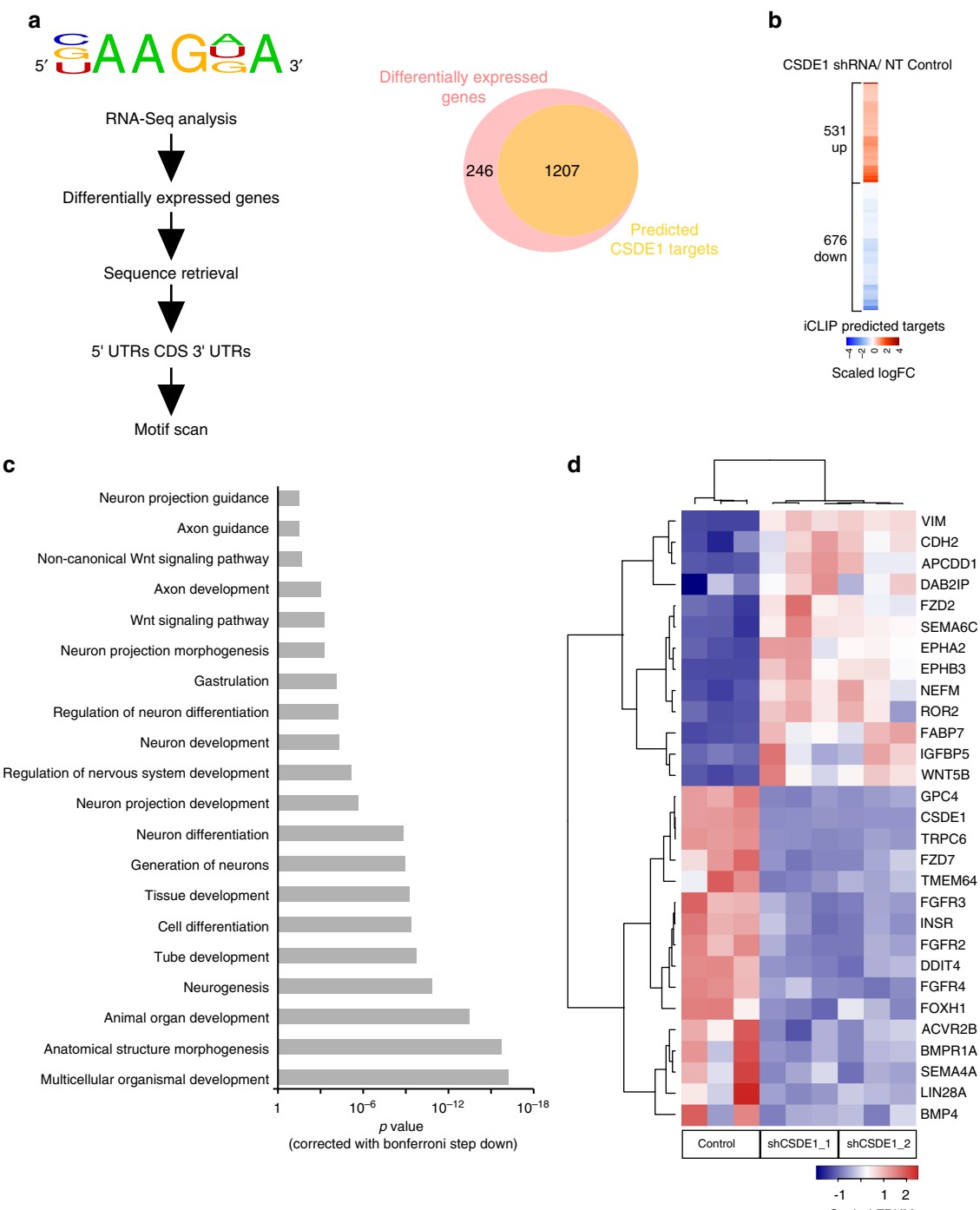

**Fig. 8** CSDE1 downregulation changes the hESC transcriptome to a neural differentiation-prone state. **a** Schematic representation of the methodology underlying CSDE1 binding motif-based searches in differentially expressed transcripts on CSDE1 KD. Venn diagram represents the number of unique transcripts shared between differentially expressed genes from CSDE1 KD H9 hESCs and predicted CSDE1 targets. Transcripts showing a $\log_2$-fold change at a False Discovery Rate (FDR) < 0.05 were retained as significantly differentially expressed. **b** Heatmap depicting the $\log_2$-fold change of the differentially expressed transcripts (FDR < 0.05) with at least one CSDE1 binding site identified in CSDE1 KD hESCs by RNA-sequencing analysis. **c** Bar graph representing the top gene ontologies (Biological Processes) of the differentially expressed transcripts with predicted CSDE1 motif in CSDE1 KD hESCs. **d** Heatmap representing a subset of candidate differentially expressed genes from RNA-sequencing analysis

defects in ventricular trabeculation and atrioventricular cushions[51].

Whereas downregulation of CSDE1 in hESCs induces their differentiation into PAX6-positive cells that lose the expression of pluripotency markers, this RBP has been shown to prevent differentiation of mESCs into primitive endoderm-like cells[20]. In the developing mouse embryo, primitive endoderm specification occurs within the inner cell mass from the mid-blastocyst stage[34]. This extraembryonic tissue is characterized by maintaining high levels of pluripotency markers (e.g., OCT4, NANOG), while expressing endoderm markers such as GATA6, GATA4, or AFP[20,34]. To explain the differences between our findings and those reported in mESCs, it is important to note the distinct pluripotent states exhibited by mESCs and hESCs in vitro[33].

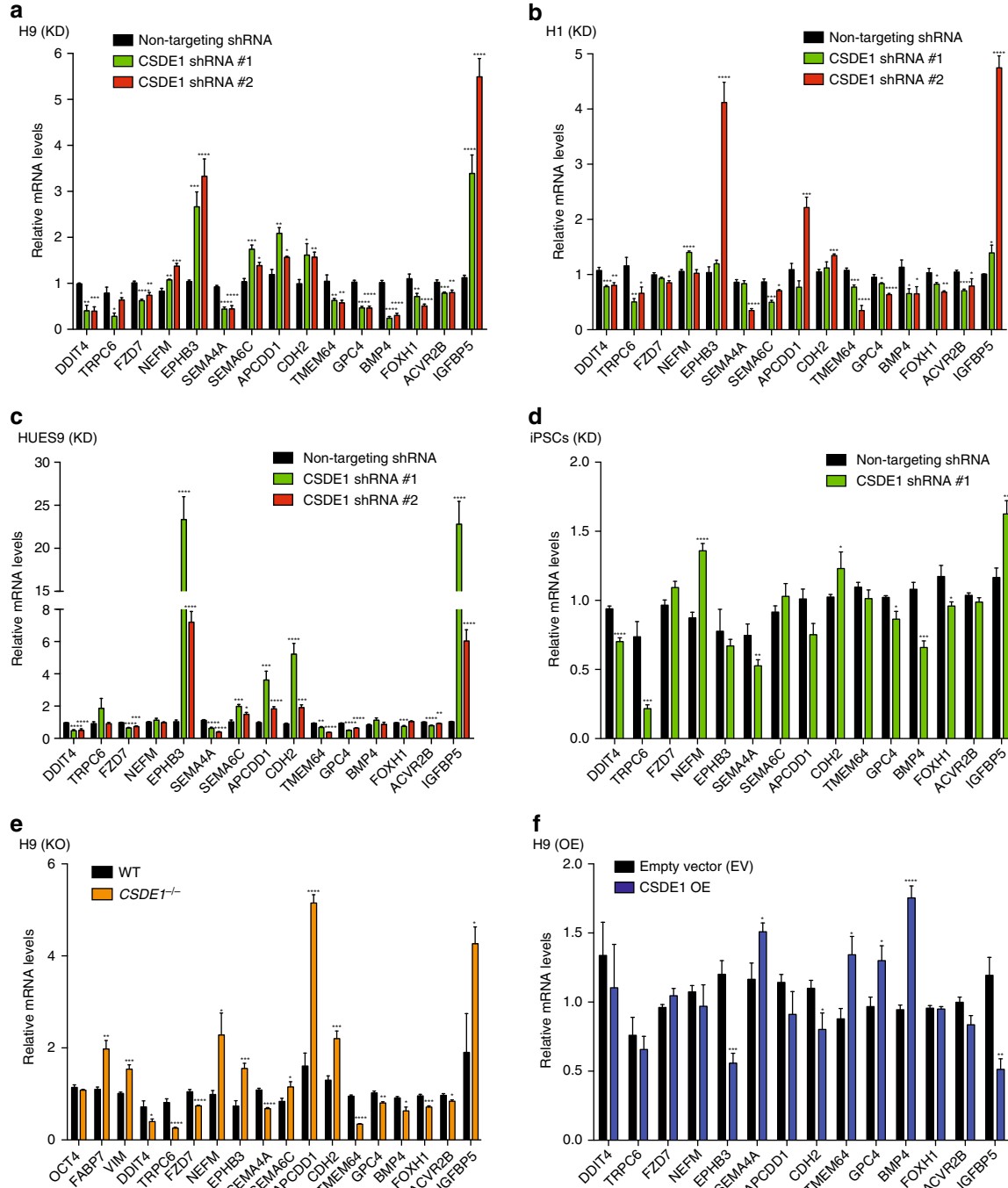

**Fig. 9** CSDE1 regulates the steady-state mRNA levels of neural factors in multiple pluripotent cell lines. **a–d** qPCR analysis of distinct CSDE1 KD hESC and iPSC lines daily monitored to remove differentiated cells. Graphs (relative expression to non-targeting (NT) shRNA) represent the mean ± s.e.m. **a** H9 hESCs, $n$ = twelve biological replicates from five independent experiments; **b** H1 hESCs, $n$ = six biological replicates from two independent experiments; **c** HUES9 hESCs, $n$ = nine biological replicates from two independent experiments; **d** iPSCs, $n$ = six biological replicates from two independent experiments. **e** qPCR analysis of $CSDE1^{-/-}$ H9 hESCs daily monitored to remove differentiated cells. Graph (relative expression to wild-type H9 hESCs) represents the mean ± s.e.m of six biological replicates from two independent experiments. **f** qPCR analysis (relative expression to EV H9 hESCs) in CSDE1 OE hESCs. Data represent two independent experiments with three biological replicates. KD = CSDE1 knockdown. KO = CSDE1 knockout. OE = CSDE1 overexpression. All the statistical comparisons were made by Student's $t$-test for unpaired samples. $P$-value: *($P < 0.05$), **($P < 0.01$), ***($P < 0.001$), **** ($P < 0.0001$)

When cultured in serum and leukemia inhibitory factor (LIF), mESCs are in a naive state resembling the pluripotent state observed in the inner cell mass of the pre-implantation embryo[33]. As such, mESC cultures are heterogeneous and consist of at least two morphologically indistinguishable cell types, resembling either primitive endoderm lineages or primed progenitors of the epiblast[52]. On the other hand, hESCs are markedly different

in vitro from mESCs and exhibit a more primed state that resembles post-implantation embryonic configurations[33]. Thus, hESCs are more similar to epiblast stem cells than mESCs and lack the ability to differentiate spontaneously into primitive endoderm. Moreover, CSDE1 can modulate different biological pathways with high versatility depending on its interaction with specific proteins and target transcripts in a dynamic process

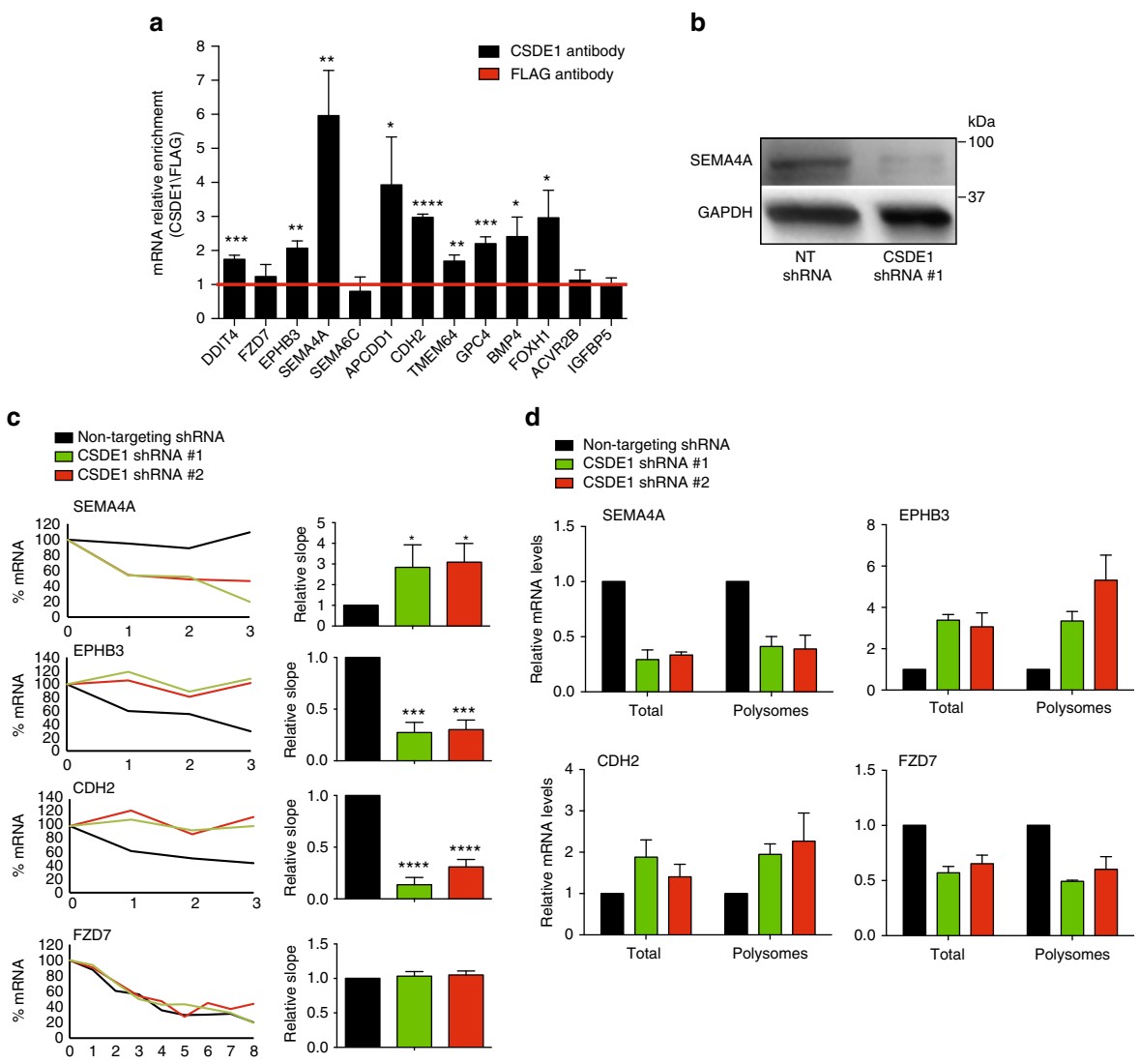

**Fig. 10** CSDE1 binds mRNAs involved in neurogenesis and post-transcriptionally regulates their steady-state levels. **a** RIP with CSDE1 antibody. Quantitative PCR analysis of the indicated genes is expressed as fold enrichment over RIP performed with FLAG control antibody. Graph (relative enrichment to FLAG antibody) represents the mean ± s.e.m. ($n = 4$ independent experiments). **b** Western blot analysis of H9 hESCs with antibody to SEMA4A. GAPDH is the loading control. **c** mRNA levels were determined after the indicated time of actinomycin D (ActD) treatment (5 µg ml$^{-1}$) by qPCR. For each gene, the right graph corresponds to a representative experiment showing the percentage of mRNA relative to time = 0. In the left panel, mRNA degradation is shown as relative slope to non-targeting shRNA hESCs (mean ± s.e.m. of four independent experiments). **d** Polysome profiling experiments followed by qPCR analysis. Total and polysome fractions mRNA levels are expressed as relative values to total and polysome fractions of NT shRNA hESCs, respectively. Graph represents the mean ± s.e.m. ($n = 4$ independent experiments). No further changes in *SEMA4A*, *EPHB3*, *CDH2*, and *FZD7* were observed in polysome fractions compared with total cell extracts. In **a** and **c** the statistical comparisons were made by Student's *t*-test for unpaired samples. In **d** the statistical comparisons were made by Student's *t*-test for paired samples (the mRNA polysome fraction was paired to its corresponding total mRNA in each independent experiment). *P*-value: *($P < 0.05$), **($P < 0.01$), ***($P < 0.001$), **** ($P < 0.0001$)

associated with the cell type and status[19–21]. Interestingly, CSDE1 destabilizes *GATA6* mRNAs in mESCs[20]. However, we did not observe changes in *GATA6* levels or other endodermal markers in hESCs. Therefore, CSDE1 could regulate distinct targets depending on the pluripotent state. In addition, mESCs and hESCs present distinct culture requirements that may impinge upon CSDE1 regulation of specific transcripts. For instance, LIF activates the JAK-STAT3 pathway and is a key ingredient for culturing mESCs in the absence of feeder cells[33]. On the contrary, hESCs and human iPSCs do not require LIF signaling whereas FGF2 and TGFß1/Activin A are core signaling pathways to maintain their pluripotent state[33]. Remarkably, we found that CSDE1 modulates key components of FGF2 and TGFß1/Activin

A pathways in hESCs. Finally, we cannot discard that the distinct phenotypes observed in hESCs and mESCs upon CSDE1 down-regulation are associated to unknown genetic differences between species. For instance, 81.5% of the CSDE1 mRNA targets identified in human melanoma do not overlap with those targets identified in *D. melanogaster*, indicating profound differences between species[16].

To define the mechanisms by which CSDE1 regulates neural differentiation and its targets in hESCs, we proposed to use undifferentiated CSDE1 KD colonies. Hence, we transferred typical undifferentiated colonies followed by extensive monitoring to remove differentiated cells. By these means, we avoided the use of CSDE1 KD heterogeneous populations that contain cells

with prominent morphological differences. These cells could present an advanced state of differentiation that may mask the original events triggered by loss of CSDE1. In our first approach, we performed quantitative proteomic analysis of undifferentiated CSDE1 KD hESCs to identify these events. Proteomic analysis revealed an upregulation of FABP7 and VIM protein levels, providing a further link between loss of CSDE1 and neurogenesis. These factors are normally expressed in neuroepithelial cells[37,39]. In fact, FABP7 is required for the maintenance of neuroepithelial cells during early embryonic development[39]. FABP7 and VIM are also highly expressed in radial glial cells. These cells are originated from neuroepithelial cells during development after the onset of neurogenesis[38]. Radial glia represent more fate-restricted progenitors than neuroepithelial cells and successively replace them[38]. A fate mapping study found that FABP7-positive radial glial cells, regardless their position in the brain, go through a neurogenesis stage giving rise to most of the neurons in the brain[28]. Radial glia not only exhibit neuroepithelial but also astroglial properties[38]. In mice, this transition occurs between embryonic day 10 (E10), when no astroglial markers can be detected, and E12, when most of the central nervous system regions are dominated by astroglial marker-positive progenitor cells[38]. Notably, our results show that loss of CSDE1 triggers FABP7 and VIM levels without changes in astrocyte markers, indicating that CSDE1 downregulation induces differentiation into NPCs without forming radial glia cells. Interestingly, this upregulation of FABP7 and VIM is required for neurogenesis of hESCs. Our results are supported by the essential role of FABP7 in brain development[29]. FABP7 is involved in long chain fatty acid uptake, transport and metabolism[29]. Long chain polyunsaturated fatty acids are enriched in developing brain and are essential for normal development of the central nervous system. Interestingly, docosahexaenoic acid (DHA), the main ligand of FABP7, promotes neurogenesis of both pluripotent and neural stem cells[53,54].

Our data indicate that CSDE1 post-transcriptionally downregulates FABP7 by modulating both its mRNA steady-state levels and translation. VIM has been recently described as a direct target of CSDE1 in melanoma[16]. In these cells, CSDE1 induces VIM protein expression with pro-oncogenic effects that contribute to melanoma invasion and metastasis. In contrast, CSDE1 preserves low levels of VIM in hESCs and contributes to sustain pluripotency. Therefore, CSDE1 has opposite effects on VIM levels depending on the cell type, a process that could be associated to the distinct mechanisms by which CSDE1 posttranscriptionally regulates VIM in these cells. We found that CSDE1 enhances VIM mRNA turnover in hESCs. In melanoma, CSDE1 promotes translation elongation of VIM without altering its steady-state transcript levels[16]. The distinct regulatory mechanisms may depend on many factors such as changes in the interaction with other RBPs and/or the transcript regions where CSDE1 primarily binds.

While proteomic analysis revealed differences in FABP7 and VIM on CSDE1 knockdown, we hypothesized that additional mechanisms could contribute to neural differentiation because of the potential of CSDE1 to bind hundreds of RNAs through its five CSDs. Notably, we identified changes in the steady-state levels of over 1000 RNAs by transcriptomic analysis. For instance, we found alterations in known regulatory factors involved in pluripotency and neural differentiation that could not be quantified by proteomics as the method was not sensitive enough to detect these proteins (e.g., DDIT4, TRPC6, EPHB3, SEMA4A, TMEM64, BMP4, FGFRs, FOXH1, ACVR2B, IGFBP5). It is also important to note that not all of the transcripts impaired in CSDE1 KD hESCs may result in changes at the protein level as compensatory translational mechanisms could mediate in this

process[16]. Nevertheless, changes at the transcript level of specific regulators might contribute to facilitate neural differentiation under specific conditions (e.g., neural induction treatment).

GOBP analysis of transcripts impaired upon CSDE1 KD revealed a strong enrichment for regulators of extracellular matrix organization, development, neuron differentiation and neuron projection guidance. Interestingly, the direct CSDE1 targets identified in melanoma cells[16] are also enriched for factors with a role in extracellular matrix organization, organ development, anatomical structure morphogenesis and neuron projection guidance (Supplementary Data 6). Although not all of the transcripts impaired upon CSDE1 knockdown are direct targets, our results suggest that loss of CSDE1 induces pro-neurogenic changes in the transcriptome landscape of hESCs and predispose them to neural differentiation. Among these changes, we found a decrease in the levels of neuron differentiation inhibitors (e.g., DDIT4) whereas neurogenic factors are increased (e.g., EPHB3). Moreover, we detected changes in WNT-signaling regulators such as GPC4, TMEM64 or FZD7, a WNT receptor required for pluripotency[41,42]. An efficient method to generate NPCs from hESCs is the use of specific inhibitors of the TGF-beta/SMAD signaling pathway, such as Noggin and SB431542, that function by blocking BMP and Activin A signaling, respectively[55]. Interestingly, loss of CSDE1 induces a decrease in the transcript levels of BMP4, the activin receptor ACVR2B and FOXH1, a transcriptional activator that forms a complex with SMAD2/SMAD4[56].

Taken together, we identify CSDE1 as a negative regulator of neural differentiation and neurogenesis. The direct connection between CSDE1 levels and cell fate decisions provides novel insights into pluripotent stem cell biology and posttranscriptional regulation of neurogenesis. Thus, a further understanding of CSDE1 modulation could provide novel therapeutic approaches for regenerative medicine and the treatment of neurodegenerative disorders.

## Methods

**hESC culture and differentiation.** The H9 (WA09) and H1 (WA01) hESC lines were obtained from WiCell Research Institute. The HUES6 and HUES9 hESC lines were obtained from Harvard Stem Cell Institute. The human iPSC line (hFIB2-iPS4) generated and fully characterized for pluripotency in ref. [57], was a gift from G.Q. Daley. hESC/iPSC lines were maintained on Geltrex (ThermoFisher Scientific) using mTeSR1 (Stem Cell Technologies). Undifferentiated colonies were passaged using dispase (2 mg ml$^{-1}$), and scraping the colonies with a glass pipette. Genetic identity of H9 and H1 hESCs was assessed by short tandem repeat (STR) analysis using the Promega PowerPlex 21 system (Promega Corporation) by Eurofins Genomics (Germany). The H9 and H1 hESC lines used in our study match exactly the known STR profile of these cells across the 8 STR loci analyzed. All the cell lines used in this study were tested for mycoplasma contamination at least once every three weeks. No mycoplasma contamination was detected. Research involving hESC lines was performed with approval of the German Federal competent authority (Robert Koch Institute).

Neural differentiation of hESCs/iPSCs was performed following the monolayer culture method with STEMdiff Neural Induction Medium (Stem Cell Technologies) based on ref. [55]. Human pluripotent stem cells were rinsed once with PBS and then we added 1 ml of Gentle Dissociation Reagent (Stem Cell Technologies) for 10 min. After this, we gently dislodged pluripotent stem cells and added 2 ml of Dulbecco's Modified Eagle Medium (DMEM)-F12 + 10 μM ROCK inhibitor (Abcam). Then, we centrifuged cells at 300×g for 10 min. Cells were resuspended on STEMdiff Neural Induction Medium + 10 μM ROCK inhibitor and plated on polyornithine (15 μg ml$^{-1}$)/laminin (10 μg ml$^{-1}$)-coated plates (200,000 cells cm$^{-2}$). Following this protocol, we were able to induce neural differentiation of H9, H1, HUES6 hESCs, as well as iPSCs. However, we were not able to induce neural differentiation of the HUES9 hESC line.

For pan-neuronal differentiation, NPCs were dissociated with Accutase (Stem Cell Technologies) and plated into neuronal differentiation medium (Dulbecco's Modified Eagle Medium (DMEM)/F12, B27, N2 (ThermoFisher Scientific), 1 μg ml$^{-1}$ laminin (ThermoFisher Scientific), 20 ng ml$^{-1}$ GDNF (Peprotech), 20 ng ml$^{-1}$ BDNF (Peprotech), 200 nM ascorbic acid (Sigma) and 1 mM dibutyryl-cyclic AMP (Sigma)) onto polyornithine/laminin-coated plates[30]. Cells were differentiated for 1 month (otherwise the time is indicated in the respective figures), with weekly feeding of neuronal differentiation medium.

Cardiomyocyte differentiation was performed as described in ref. [58]. Confluent H1 hESCs were dissociated into single cells with Accutase at 37 °C during 10 min followed by inactivation using two volumes of DMEM/F12. Cells were counted and 230,000 cells/cm², where plated in ITS medium (Corning), containing 1.25 μM CHIR 99021 (AxonMedchem) and 1.25 ng ml⁻¹ BMP4 (R&D), and seeded on Matrigel-coated 24-well plates. After 24 h, medium was changed to transferrin/selenium (TS) medium. After 48 h, medium was changed to TS medium supplemented with 10 μM canonical Wnt-Inhibitor IWP-2 (Santa Cruz) for 48 h. Then, medium was changed to fresh TS until beating cells were observed at days 8 to 10. Finally, medium was changed to KnockOut DMEM (ThermoFisher Scientific) supplemented with 2% FCS, L-Glutamine and Penicillin/Streptomycin until cells were used for downstream analysis. Endoderm differentiation of H9 hESCs was performed using STEMdiff Definitive Endoderm Kit (Stem Cell Technologies).

**mESC culture and differentiation**. The E14 mESC line was maintained in the naive state on gelatin-coated plates in Knockout™ DMEM (ThermoFisher Scientific) with 15% Hyclone competent serum (VWR), 2 mM L-glutamine, 1% penicillin/streptomycin (Biochrom), 1 mM sodium pyruvate, 0.1 mM MEM non-essential amino acids, and 0.1 mM β-mercaptoethanol (all are from ThermoFisher Scientific) supplemented with LIF (1000 U; Merck) and 2i (1 μM PD0325901 and 3 μM CHIR99021; Miltenyi Biotech).

E14 mESCs were also adapted to serum-free medium containing a 1:1 mixture of advanced Dulbecco's modified Eagle's medium F12 and neurobasal medium (ThermoFisher Scientific), supplemented with 1 × N2, 1 × B27, and 40 mg ml⁻¹ BSA (ThermoFisher Scientific) plus 2 mM L-glutamine, 1% penicillin/streptomycin, 0.1 mM β-mercaptoethanol, and 12.5 μg ml⁻¹ insulin (Sigma-Aldrich), including LIF and 2i (Fig. 2b). Differentiation of E14 mESCs to NPCs was performed in serum-free medium on matrigel-coated (BD Biosciences) plastic dishes following the protocol described in ref. [59]. Briefly, E14 mESCs were cultured without the inhibitors and with 10 ng ml⁻¹ basic fibroblast growth factor (bFGF; ThermoFisher Scientific) for two days, with a combination of bFGF and 5 μM XAV (Sigma-Aldrich) for 1 day, and with XAV alone for 2 days. After six days of culture, both naive mESCs and NPCs were collected for western blotting analysis (Fig. 2a).

**Collection of mouse embryos**. Wild-type FVB/N mice were setup for timed mating and the embryos collected at embryonic day E8.5. The neural-rich anterior region was cut before the heart level and the streak-rich posterior region was cut after the last pair of somites. Tissues from two embryos were pooled for Western blot analysis. Experiments involving sacrifice of wild-type animals to obtain E8.5 embryos were approved by local government authorities (Landesamt für Natur, Umwelt und Verbraucherschutz Nordrhein-Westfalen, Germany).

**Lentiviral vectors**. Lentivirus (LV)-non targeting shRNA control, LV-CSDE1shRNA #1 (TRCN0000364597), LV-CSDE1 shRNA #2 (TRCN0000364598), LV-CSDE1 shRNA #3 (TRCN0000364674), LV-FABP7 shRNA #1 (TRCN0000059743), LV-FABP7 shRNA #2 (TRCN0000059745), LV-VIM shRNA #1 (TRCN0000029119), LV-VIM shRNA #2 (TRCN0000029121), LV-YBX1 shRNA (TRCN0000315307), LV-YBX2 shRNA (TRCN0000107505), LV-YBX3 shRNA (TRCN0000297824), LV-EIF1AX shRNA #1 (TRCN0000299430), LV-EIF1AX shRNA #2 (TRCN0000062621), LV-EIF2A shRNA (TRCN0000143559), LV-EIF5A shRNA #1 (TRCN0000062552), LV-EIF5A shRNA #2 (TRCN0000062551), LV-DIS3 shRNA (TRCN0000049841) and LV-EXOSC3 shRNA (TRCN0000050408) in pLKO.1-puro vector were obtained from Mission shRNA (Sigma).

CSDE1-overexpressing lentiviral construct (CSDE1(OE)) was generated as follows. Human *CSDE1* complementary DNA was PCR-amplified and cloned intoCD522A-1 pCDH cDNA Cloning Lentivector (System Biosciences) using NheI and BamHI. This construct was sequence verified and thereafter transfected into packaging cells to produce high titer lentiviruses. CD522A-1 pCDH contains MSCV CpG-deficient promoter incorporated into the 3′HIV LTR for durable overexpression of a target gene in ESCs. The MSCV is the 5′-LTR promoter of murine stem cell virus. After integration into genomic DNA, the hybrid HIV/MSCV promoter provides stable overexpression of the target gene as well as puromycin resistance gene. Moreover, CpG mutations in the MSCV LTR reduce transcriptional silencing in ESCs[60].

**Lentiviral infection of hESCs**. Transient infection experiments for shRNA screen were performed as follows. hESC colonies growing on Geltrex were incubated with mTesR1 medium containing 10 μM ROCK inhibitor (Abcam) for 2 h and individualized using Accutase. Hundred thousand cells were plated on Geltrex plates and incubated with mTesR1 medium containing 10 μM ROCK inhibitor for 1 day. Then, cells were infected with 5 μl of concentrated lentivirus. Plates were centrifuged at 800g for 1 h at 30 °C. Cells were fed with fresh media the day after to remove virus. After 1 day, cells were selected for lentiviral integration using 2 μg ml⁻¹ puromycin (ThermoFisher Scientific). Cells were then collected for qPCR experiments after 4–6 days of infection.

For FABP7, VIM and CSDE1 shRNA experiments we generated stable transfected hESCs. To obtain shRNA stable lines, hESC colonies growing on Geltrex were incubated with mTesR1 medium containing 10 μM ROCK inhibitor for 1 h and individualized using Accutase. Fifty thousand cells were infected with 20 μl of concentrated lentivirus in the presence of 10 μM ROCK inhibitor for 1 h. Cell suspension was centrifuged to remove virus, passed through a mesh of 40 μM to obtain individual cells, and plated back on a feeder layer of mitotically inactive mouse embryonic fibroblasts (MEFs) in hESC media (DMEM/F12, 20% knockout serum replacement (ThermoFisher Scientific), 0.1 mM non-essential amino acids, 1 mM L-glutamine, β-mercaptoethanol and 10 ng ml⁻¹ bFGF (Joint Protein Central)) supplemented with 10 μM ROCK inhibitor. After a few days in culture, small hESC colonies arose. Then, we performed 1 μg ml⁻¹ puromycin selection during 2 days and colonies were manually passaged onto fresh MEFs to establish new hESC lines. Following this protocol, we also generated CSDE1 (OE) hESC stable lines.

**Generation of *CSDE1⁻/⁻* hESCs by CRISPR/Cas9 system**. *CSDE1* third exon sequence was obtained from ENSEMBL Genome browser. Two guide sequences (Guide A forward: CAGCAGCATTAACATCACC, and reverse: ACCA-CACTTTGAAAACCAC; Guide B forward: TTCACCAGTTTAACAGCAA, and reverse: TCCCTGAAGAACGAATGAA) targeting this exon were generated using the Zhang lab online resource (http://crispr.mit.edu/). Guide-carrying plasmids were designed using Cas9-puromycin selection plasmid (pX335-U6-Chimeric_BB-CBh-hSpCas9n, Addgene)[61]. H9 hESCs were transfected with the guide-carrying plasmid using FuGene HD (Promega). 24 h after the transfection, 0.5 μg ml⁻¹ puromycin selection was performed for 24 h followed by maintenance of hESCs with mTeSR1 media. Single cell split was performed prior to colony pick for genotyping. DNA isolation was done using QuickExtract (Epicentre). PCR for *CSDE1* third exon was performed (Forward: TTGTTTTGGTTAATCCTCATGGCA, and Reverse: AGCTCTCTTTCGTGCAAACTGA) to identify *CSDE1⁻/⁻* hESCs.

**Sample preparation for quantitative proteomics and analysis**. For the comparison between H9 hESCs and their differentiated neuronal counterparts (Supplementary Table 1 and Supplementary Data 1), we performed tandem mass tag (TMT) proteomics. 50 μg of protein were precipitated with 23% TCA. Proteins were solubilized in 100 μl of 100 mM TEAB and processed according to TMTsixplex Isobaric Mass tag kit protocol (Thermo Scientific, catalog #90064). MudPIT analysis was performed with an Agilent 1100 G1311 quaternary pump and a Thermo LTQ-Orbitrap Elite using an in-house built electrospray stage. Protein/peptide identification and protein quantitation were done with Integrated Proteomics Pipeline - IP2 (Integrated Proteomics Applications, Inc., San Diego, CA. http://www.integratedproteomics.com/). Tandem mass spectra were extracted from raw files using RawExtract 1.9.9[62] and then searched against a Uniprot human database with reversed sequences using ProLuCID[62,63]. The search space included all fully-tryptic peptide candidates. Peptide candidates were filtered using DTASelect, with the following parameters: -p 2 -y 0 --trypstat --fp .05 --extra --pI -DM 10 --DB --dm -in -t 1 --brief --quiet[62]. Quantitation was performed using Census[64]. Statistical comparisons were made by Student's t-test. False Discovery Rate (FDR) adjusted p-value (q-value) was calculated using the Benjamini–Hochberg procedure.

In Fig. 1a, we analyzed CSDE1 protein levels using available quantitative proteomics data comparing hESCs with their NPC and neuronal counterparts[30]. The analysis was carried out on LFQ values, which were subjected to the variance stabilization transformation method (limma). CSDE1 levels were examined by linear modeling including cell type and experimental batch as variable using limma's moderated t-statistics framework.

For the comparison between control and CSDE1 KD H9 hESCs (Supplementary Table 3 and Supplementary Data 3), we performed label-free quantitative (LFQ) proteomics. Cells were collected in urea buffer (8 M urea, 50 mM ammonium bicarbonate and 1x complete protease inhibitor mix with EDTA (Roche)), homogenized with a syringe and cleared using centrifugation (16,000 g, 20 min). Supernatants were reduced (1 mM DTT, 30 min), alkylated (5 mM iodoacetamide (IAA), 45 min) and digested with trypsin at a 1:100 w/w ratio after diluting urea concentration to 2 M. One day after, samples were cleared (16,000 g, 20 min) and supernatant was acidified. Peptides were cleaned up using stage tip extraction[65]. The liquid chromatography tandem mass spectrometry (LC-MS/MS) equipment consisted out of an EASY nLC 1000 coupled to the quadrupole based QExactive instrument (Thermo Scientific) via a nano-spray electroionization source. Peptides were separated on an in-house packed 50 cm column (1.9 μm C18 beads, Dr. Maisch) using a binary buffer system: A) 0.1% formic acid and B) 0.1 % formic acid in ACN. The content of buffer B was raised from 7 % to 23 % within 120 min and followed by an increase to 45 % within 10 min. Then, within 5 min buffer B fraction was raised to 80 % and held for further 5 min after which it was decreased to 5 % within 2 min and held there for further 3 min before the next sample was loaded on the column. Eluting peptides were ionized by an applied voltage of 2.2 kV. The capillary temperature was 275 °C and the S-lens RF level was set to 60. MS1 spectra were acquired using a resolution of 70,000 (at 200 m/z), an Automatic Gain Control (AGC) target of 3e6 and a maximum injection time of 20 ms in a scan range of 300–1750 Th. In a data dependent mode, the 10 most intense peaks were selected for isolation and fragmentation in the HCD cell using a normalized collision energy of 25 at an isolation window of 2.1 Th. Dynamic exclusion was

enabled and set to 20 s. The MS/MS scan properties were: 17,500 resolution at 200 m/z, an AGC target of 5e5 and a maximum injection time of 60 ms. All label-free proteomics data sets were analyzed with the MaxQuant software (release 1.5.3.8). We employed the LFQ mode[66] and used MaxQuant default settings for protein identification and LFQ quantification. All downstream analyzes were carried out on LFQ values with Perseus (v. 1.5.2.4)[67].

**Protein immunoprecipitation for interactome analysis.** hESCs were lysed in modified RIPA buffer (50 mM Tris-HCl (pH 7.4), 150 mM NaCl, 1% IgPal, 0.25% sodium deoxycholate, 1 mM EDTA, 1 mM PMSF) supplemented with protease inhibitor (Roche). Lysates were centrifuged at 10,000 g for 10 min at 4 °C. Then, the supernatant was collected and incubated with CSDE1 antibody (Abcam, #176584, 1:100) for 30 min and subsequently with 100 µl Protein A beads (Miltenyi) for 1 h on the overhead shaker at 4 °C. As a control, the same amount of protein was incubated with anti-FLAG antibody (SIGMA, F7425, 1:100) in parallel. After this incubation, supernatants were subjected to magnetic column purification. Three washes were performed using wash buffer 1 (containing 50 mM Tris–HCl (pH 7.4), 150 mM NaCl, 5% glycerol and 0.05% IgPal). Next, columns were washed five times with wash buffer 2 (containing 50 mM Tris–HCl (pH 7.4), 150 mM NaCl). Then, columns were subjected to in-column tryptic digestion containing 7.5 mM ammonium bicarbonate, 2 M urea, 1 mM DTT and 5 ng ml$^{-1}$ trypsin. Digested peptides were eluted using two times 50 µl of elution buffer 1 containing 2 M urea, 7.5 mM Ambic, and 5 mM IAA. Digests were incubated over night at room temperature with mild shaking in the dark. Samples were stage-tipped the next day for label-free quantitative proteomics and analyzed with MaxQuant software. The downstream analyzes were carried out on LFQ values with Perseus (v. 1.5.2.4).

**Western blot.** Cells were scraped from tissue culture plates and lysed in protein cell lysis buffer (10 mM Tris–HCl, pH 7.4, 150 mM NaCl, 10 mM EDTA, 50 mM NaF, 1% Triton X-100, 0.1% SDS supplemented with 20 µg ml$^{-1}$ Aprotinin, 2 mM sodium orthovanadate, 1 mM phenylmethylsulphonyl fluoride and protease inhibitor (Roche)) by incubating samples for 10 min on ice and homogenization through syringe needle (27G). The, cell lysates were centrifuged at 10,000×g for 10 min at 4 °C and the supernatant was collected. Protein concentrations were determined with a standard BCA protein assay (Thermoscientific). Approximately 20–30 µg of total protein was separated by SDS–PAGE, transferred to nitrocellulose membranes (Millipore) and subjected to immunoblotting. Western blot analysis was performed with anti-PAX6 (Stem Cell Technologies, #60094, 1:200), anti-OCT4 (Stem Cell Technologies, #60093, 1:500), anti-SOX2 (Abcam, #97959, 1:1,000), anti-BLBP (Abcam, #32423, 1:500), anti-VIM (Abcam, #92547, 1:1,000), anti-RPL7 (Genetex, #114727, 1:1,500), anti-RPS27 (Proteintech, 15355-1-AP, 1:500), anti-SEMA4A (Proteintech, #12288-2-AP, 1:500), anti-Nestin (Stem Cell Technologies, #60091, 1:1,000), anti-GFAP (Merck Millipore, AB5804, 1:1,000), anti-ß-actin (Abcam, #8226, 1:1,000) and anti-GAPDH (Abcam, #8226, 1:3,000). Analysis of CSDE1 levels were performed with anti-CSDE1 (Abcam, #9484, 1:2,000) for human cells and with anti-CSDE1 (Abcam, #201688, 1:1,000) for mouse cells. Uncropped versions of western blots are presented in Supplementary Fig. 21.

**Immunocytochemistry.** Human cells were fixed with paraformaldehyde (4% in PBS) for 30 min, followed by permeabilization (0.2% Triton X-100 in PBS for 10 min) and blocking (3% BSA in 0.2% Triton X-100 in PBS for 10 min). Human cells were incubated in primary antibody for 2 h at room temperature (Rabbit anti-CSDE1 (Abcam, #176584, 1:100), Mouse anti-OCT4 (Stem Cell Technologies, #60093, 1:200), Rabbit anti-PAX6 (Stem Cell Technologies, #60094, 1:300), Mouse anti-MAP2 (Sigma, #1406, 1:200) and Rabbit anti-GFAP (Millipore, AB5804, 1:500)). Then, cells were washed with 0.2% Triton-X/PBS and incubated with secondary antibody (Alexa Fluor 488 goat anti-mouse (Thermo Fisher Scientific, A-11029, 1:500), Alexa Fluor 568 goat anti-rabbit (Thermo Fisher Scientific, A-11011, 1:500), and 2 µg ml$^{-1}$ Hoechst 33342 (Life Technologies, #1656104) for 1 h at room temperature. 0.2% Triton-X/PBS and distilled water wash were followed before the cover slips were mounted.

For immunofluorescence of mESCs, we fixed the cells in 4% paraformaldehyde at room temperature for 10 min and rinsed with PBS. Before adding primary antibodies, the mESCs were additionally fixed in methanol at −20 °C for 10 min, permeabilized with 0.5% Triton X-100/PBS for 5 min at room temperature, and blocked in 5% heat-inactivated goat serum (ThermoFisher Scientific) for 15 min at room temperature. The mESCs were then incubated in rabbit anti-CSDE1 (Abcam, #201688, 1:300) and rat anti-Nanog (eBioscience, #14–5761, 1:200). Afterwards, the mESCs were washed with 0.2% Triton X-100/PBS and incubated with secondary antibodies Alexa Fluor 488 goat anti-rat (ThermoFisher Scientific, A-11006, 1:1000), Alexa Fluor 568 goat anti-rabbit (ThermoFisher Scientific, A11011, 1:1000), and DAPI (AppliChem, A4099, 1:1000) for 1 h at room temperature. Finally, the mESCs were washed with 0.2% Triton X-100/PBS and distilled water, and cover slips were mounted using ProLong Gold anti-fade reagent (New England Biolabs).

**Alkaline phosphatase assay.** Cells were washed with PBS, fixed in 100% methanol for 10 min and air dried. Then, cells were incubated with the staining

solution (mixture of 2 mg ml$^{-1}$ Napthol AS-MX phosphate (Sigma) in 0.1 M Tris–HCl pH 9.2 and 1 mg ml$^{-1}$ Fast Red TR saltTM (Sigma) in 0.1 M Tris–HCl pH 9.2 to 1:10 dilution) for 10–15 min in dark. The reaction was stopped reaction by rinsing cells twice with distilled water.

**Bromodeoxyuridine (BrdU) proliferation assay.** hESCs were incubated with media containing 15 µM ml$^{-1}$ BrdU for 24 h. Cells were fixed with formaldehyde 4% in PBS. Then, cells were permeabilized with 0.2% Triton X-100 in PBS for 10 min and blocked with 3% BSA-PBS for 1 h at room temperature. 2N HCl was added for 15 min at room temperature. After this, cells were incubated in 0.1 M sodium tetra-borate for 15 min at room temperature. We performed overnight incubation with rabbit anti-BrdU (ABD Serotech, 1:200) at 4 °C followed by incubation with a biotinylated anti-rabbit secondary antibody (Vector) for 2 h at room temperature. Finally, hESCs were incubated with streptavidin-AlexaFluor 568 (Jackson Immuno Research, 1:500) for 1 h. Hoechst 33342 was used to visualize nuclei.

**RNA isolation and quantitative RT-PCR.** For human cell samples, total RNA was extracted using RNAbee (Tel-Test Inc.). cDNA was generated using qScript Flex cDNA synthesis kit (Quantabio). SybrGreen real-time qPCR experiments were performed with a 1:20 dilution of cDNA using a CFC384 Real-Time System (Bio-Rad) following the manufacturer's instructions. Data were analyzed with the comparative 2ΔΔ$C_t$ method using the geometric mean of *ACTB* and *GAPDH* as housekeeping genes. See Supplementary Data 8 for details about the primers used for this assay.

**RNA sequencing.** Total RNA was extracted using RNAbee (Tel-Test Inc.). Libraries were prepared using the TruSeq Stranded mRNA Library Prep Kit. Library preparation started with 1 µg total RNA. After selection (using poly-T oligo-attached magnetic beads), mRNA was purified and fragmented using divalent cations under elevated temperature. The RNA fragments underwent reverse transcription using random primers followed by second strand cDNA synthesis with DNA Polymerase I and RNase H. After end repair and A-tailing, indexing adapters were ligated. The products were then purified and amplified (20 µl template, 14 PCR cycles) to create the final cDNA libraries. After library validation and quantification (Agilent 2100 Bioanalyzer), equimolar amounts of library were pooled. The pool was quantified by using the Peqlab KAPA Library Quantification Kit and the Applied Biosystems 7900HT Sequence Detection System. The pool was sequenced on an Illumina HiSeq 4000 sequencer with a paired- end (2 × 75bp) protocol. We used the human genome sequence and annotation (EnsEMBL 79) together with the splice-aware STAR read aligner[68] (release 2.5.1b) to map and assemble our reference transcriptome. Subsequent transcriptome analyzes on differential gene and transcript abundance were carried out with the cufflinks package[69] cuffdiff program (version 2.2.1). Transcripts showing a log$_2$-fold change at a FDR < 0.05 were retained as significantly differentially expressed. Supplementary Data 5 provides the statistical analysis of the transcriptome data.

5′ UTR, coding region and 3′ UTR sequences of all the differentially expressed genes were extracted from Ensembl BioMart. The consensus CSDE1 binding motif identified in ref. [16] was used as a query to scan these sequences using "RSAT: Regulatory Sequence Analysis tools". This analysis identified transcripts that show > 1 hit in either 5′ UTR, coding and 3′ UTR sequences. Further data analysis and graphs plotting (Fig. 8) was performed using R stats packages. Gene ontology of the differentially expressed genes was performed in Cytoscape using ClueGO App.

**RNA immunoprecipitation.** RIP experiments with anti-CSDE1 (Abcam, #176584, 1:100) were performed following the protocol described in ref. [70] with some modifications. Cells were pelleted by centrifugation at 1000×g for 10 min at 4 °C and washed several times with ice cold PBS. The final cell pellet was resuspended with an equal volume of polysome lysis buffer (100 mM KCl, 5 mM MgCl$_2$, 10 mM HEPES (pH 7.0), 0.5% NP40 (Sigma)) supplemented with 1 mM DTT, 100 per units RNase Out (Invitrogen), 400 µM VRC (New England BioLabs), and protease inhibitor. Cell lysates were incubated on ice for 10 min and homogenized through syringe needle. Antibody coating of protein A beads was prepared by pre-swelling protein-A sepharose beads (Thermoscientific) in NT2 buffer (50 mM Tris–HCl (pH 7.4), 150 mM NaCl, 1 mM MgCl$_2$, 0.05% NP40) supplemented with 5% BSA to a final ratio of 1:5 for at least 1 h at 4 °C prior to use. Then, the antibody was added to bead slurry and incubated for 2 h at 4 °C. Immediately before use, antibody-coated beads were washed (five times) with ice cold NT2 buffer followed by resuspension of beads in ice cold NT2 buffer supplemented with 200 units of an RNase inhibitor, 400 µM vanadyl ribonucleoside complexes, 1 mM DTT, and 20 mM EDTA. The cell lysate was mixed with antibody-coated beads and incubated 2 h at room temperature. The beads were washed five times with ice-cold NT2 buffer and finally washed with NT2 buffer with 1% Triton X-100. RNA extraction was done from the immunoprecipitated pellet.

**CSDE1 pulldown assay with VIM and FABP7 RNA probes.** Pull-down assay was performed using in vitro transcribed biotinylated VIM, FABP7 and 5′UTR msl2 RNA probes (negative control) according to Pierce™ Magnetic RNA-Protein Pull-Down Kit protocol with some modifications. Labeled RNA was captured using 100

μl of streptavidin magnetic beads in RNA Capture Buffer for 30 min at room temperature. Beads were washed twice in 20 mM Tris (pH 7.5), once in Protein-RNA Binding Buffer and 200 μl of H9 hESCs extract was added (8 μg μl$^{-1}$ of total protein). Samples were incubated for 30 min at room temperature, irradiated or not with 0.15 J cm $-$ 2 at 254-nm UV light, washed three times with Wash Buffer and eluted after 30 min of incubation at 37 °C with alternative Elution Buffer (Tris–HCl pH 7.4 10 mM, MgCl2 1 mM, NaCl 40 mM) with 2 μl of RNAse cocktail (Ambion AM2286; RNases A and T1). RNA pull-down specificity was assessed by Western blotting using anti-CSDE1 antibody (Abcam, #96124, 1:1,000), anti-CIRBP antibody (Abcam, #94999, 1:500), anti-CELF1 (Santa Cruz, c-20003, 1:500) and anti-Actin (Sigma, #A2066, 1:1,000).

**mRNA stability experiments.** hESCs were treated with 5 μg ml$^{-1}$ actinomycin D (Sigma). Then, cells were collected at the indicated time points for RNA extraction followed by qPCR analysis as described above.

**Ribosome fractionation.** 10% (w/v) to 50% (w/v) sucrose gradients were prepared from 10 and 50 % sucrose solutions (20 mM Tris-HCl (pH 7.4), 100 mM NaCl, 10 mM MgCl$_2$) supplemented with protease inhibitor (Roche), 20 mM DTT (Sigma), and 0.1 mg ml$^{-1}$ cyclohexamide (Sigma) in 11 ml ultracentrifuge poly-allomer tubes (Beckman Coulter) using gradient maker (Biocomp). For fractionation, hESCs were lysed by incubation on ice 10 min with polysome extraction buffer (20 mM Tris–HCl (pH 7.4), 100 mM KCl, 10 mM MgCl$_2$, 1% Triton X-100) supplemented with 25 U ml$^{-1}$ DNase I, protease inhibitor, 20 mM DTT, and 0.1 mg ml$^{-1}$ cyclohexamide followed by homogenization through syringe. Lysates were centrifuged at 10,000×g for 10 min at 4 °C. The supernatant was collected, loaded on the gradient and centrifuged at 38,000 rpm for 2 h at 4 °C using SW-41 rotor and an ultracentrifuge (Beckman Coulter). The gradients were fractionated in equal volume while absorbance at 254 nm was recorded by fraction collector (Teledyne ISCO). RNA and protein were extracted from each fraction. RNA extraction was performed as mentioned in the protocol above. Protein was obtained by Trichloroacetic acid (Sigma) precipitation from equal volume of each fraction.

**Data availability.** Transcriptome data have been deposited in the Sequence Read Archive (SRA) under the accession code SRP117243. The mass spectrometry proteomics data showed in Supplementary Table 1 have been deposited to the ProteomeXchange Consortium via the PRIDE partner repository under accession code PXD007738. The proteomics data showed in Supplementary Tables 2 and 3 are available via ProteomeXchange with the dataset identifiers PXD007271 and PXD007270, respectively. All the other data are also available from the corresponding author upon request.

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

## Acknowledgements

The Deutsche Forschungsgemeinschaft (DFG) (CECAD) and the European Research Council (ERC Starting Grant-677427 StemProteostasis) supported this research. J.J.M. and J.R.Y. were supported by the National Center for Research Resources (5P41RR011823). We thank I.S. for advice on CRISPR/Cas9 method, M.R. for analysis of enriched GO terms in interactome experiments and S.L. for her technical support. We thank the CECAD Proteomics Facility and the Cologne Center for Genomics (CCG) for their contribution and advice in proteomics and RNA sequencing experiments, respectively.

## Author contributions

H.J.L. performed most of the experiments, data analysis and interpretation through discussions with D.V. D.B. performed mesoderm differentiation and contributed to ribosome fractioning and RIP experiments. C.X. carried out mouse embryo experiments. S.G. performed CSDE1 pulldown experiments with RNA probes. G.A. performed CSDE1 binding motif and GO term analysis. C.S. helped with and performed some of the experiments. J.M. and J.R.Y. 3rd carried out TMT proteomics experiments and data analysis. F.G. contributed with her knowledge on CSDE1 function and provided critical advice for the project. H.B. contributed with his knowledge of mouse development. C.D. performed bioinformatic analysis of transcriptomics data. L.K. contributed with his knowledge of post-transcriptional regulatory mechanisms and provided critical advice for the project. D.V. planned and supervised the project. The manuscript was written by D.V. All the authors discussed the results and commented on the manuscript.

## Additional information

**Competing interests:** The authors declare no competing financial interests.

