## [Peer Review File · Nature Communications]

Reviewers' comments:

Reviewer #1 :

Lee et al performed a knockdown screen for CSD-containing RNA binding proteins controlling human stem cells. They report that CSDE1 is important for the prevention of neurogenic differentiation of ESCs. Knockdown of CSDE1 leads to increased levels of VIM and FABP7 mRNAs in ESCs that precedes differentiation into neurons. Conversely, downregulation of FABP7 or overexpression of CSDE1 leads to impaired neuronal differentiation. The study concludes that downregulation of CSDE1 during development is necessary for the specification of neuronal identity. This claim needs to be supported by a stronger quantitative evidence and extended to more than one stem cell line. Considering the difficulty of assessing the levels of CSDE1 in the developing human embryos, it would be informative to complement the in vitro work with analysis of CSDE1 in the developing mouse embryonic tissues. In the absence of this information, it remains questionable whether the mechanism is actually involved in the control of neurogenesis or whether the study revealed an interesting epiphenomenon.

Specific points:

While the authors state that they repeated some experiments in a second cell line, many of these are shown only qualitatively (Fig S2A) or not at all. Confirmation of the key findings in an independent ESC cell line is of high importance. Differentiation potential of individual human ESC lines is highly variable and therefore it is essential to establish the universality of the observations by probing several cell lines.

Downregulation of CSDE1 is not clearly demonstrated and needs to be supported by more than qualitative western blot analysis. The loading is uneven and no quantifications has been provided. The levels of mRNA need to be quantified and expression in individual cells should be assessed by immunostaining. Furthermore, existing expression profiling data on neural differentiation of ESCs should be analyzed for changes in CSDE1 expression in these independent global datasets. Demonstrating the developmental regulation of CSDE1 is a key argument for its biological relevance.

The authors reported co-IP experiments to identify CSDE1 interactors. They list the identified proteins, but do not provide any data that would control for the quality and specificity of IPs. The primary data supporting the IP experiments should be added to supplemental information.

Finally CSDE1/UNR protein has been studied in detail in cancer. Previous study reported Vimentin as a target of CSDE1 and expression analysis as well as CLIP identification of CSDE1 targets has been performed. The authors need to properly reference the published papers and use existing data to perform comparative analysis. CSDE1 was also knocked out in mouse ES cells and no defect in ES cell proliferation has been noted. This deserves to be discussed in the manuscript.

Reviewer #2 :

Lee et al explore the function of CSDE1, an RNA binding protein, in the regulation of hESC self-renewal and neuronal differentiation. They test knockdown of CSDE1 in ES cells and suggest that this results in an accelerated neural induction and neurogenesis; this is indicated by increased FABP7 and vimentin (a radial glia/neural stem cell marker). They propose a functionally important role for both FABP7 (also known as BLBP) in control of neurogenesis and that CSDE1 controls the levels through a posttranscriptional mechanism. Gain of function of CSDE1 can block neuronal differentiation. This is a novel observation.

CSDE1 emerged as a candidate regulator of hESCs from an initial proteomics and shRNA

knockdown screen. Figure 1 present the characterisation of CSDE1 protein and mRNA levels. The western is not particularly convincing that the levels in neurons are decreased, as the loading control is also much lower. There are indications of increased neural induction, but they need to better confirm knockdown and rescue the phenotype to ensure this is not an off target effect; obviously CRISPR deletion is feasible and has become rather standard definitive proof of function. They only present data using two cell lines, so limited insights are gained. Overall I am not convinced, as much more in depth analysis of the phenotype is needed.

While there are some interesting preliminary observations, the characterisation of the shRNA knockdown ES cells is rather superficial and premature. I believe this is premature for publication and unfortunately is some way away from being a significant enough body work with broad enough interest for Nature Communications. There is also no attempt to really integrate this CSDE1 function with the vast body of known regulators that control ES cell neural induction.

Reviewer #3 :

This paper by Lee et al. investigated the role of CSDE1, an RNA binding protein with multiple cold shock domains, in maintenance of hESC pluripotency. The authors found knockdown of CSDE1 led to the spontaneous differentiation of hESCs to neuronal lineage, but not astroglial lineage. The authors further showed that CSDE1 KD led to increase of FABP7 and VIM, two novel CSDE1 targets that have been implicated in neuronal development, at the mRNA and/or protein levels, probably through direct protein-RNA interaction. Overall, the paper is well written, and the results add our understanding of the contribution of RBPs in stem cell biology.

Specific comments:

A prior study (ref 35) investigated knockout of CSDE1 in mouse ESCs and found that loss of CSDE1 led to differentiation into the endoderm lineage, which is in contrast to this study. While the paper was cited in discussion, the discussion was somewhat vague and focused on expression of markers. It will be helpful to clearly describe the previous study in introduction. This will be helpful to understand why markers of different germ layers were examined. Potential factors contributing to the discrepancy can also be discussed.

The evidence of direct regulation of FABP7 and VIM by CSDE1 is moderate. Not to mention RIP is probably not the ideal assay to determine the direct protein-RNA interactions, as compared to other methods such as CLIP or mutagenesis of specific sites, the fold enrichment by RIP is moderate (4-5 fold), especially given that all tested substrates showed some extent of enrichment (which one is the negative control?).

The heterogeneity of the colonies upon CSDE1 KD complicates the interpretation of the results. This issue is noted, but it might worth to elaborate more in discussion.

Reviewer #1 (hESC and neuronal differentiation expert):

Lee et al performed a knockdown screen for CSD-containing RNA binding proteins controlling human stem cells. They report that CSDE1 is important for the prevention of neurogenic differentiation of ESCs. Knockdown of CSDE1 leads to increased levels of VIM and FABP7 mRNAs in ESCs that precedes differentiation into neurons. Conversely, downregulation of FABP7 or overexpression of CSDE1 leads to impaired neuronal differentiation. The study concludes that downregulation of CSDE1 during development is necessary for the specification of neuronal identity. This claim needs to be supported by a stronger quantitative evidence and extended to more than one stem cell line. Considering the difficulty of assessing the levels of CSDE1 in the developing human embryos, it would be informative to complement the in vitro work with analysis of CSDE1 in the developing mouse embryonic tissues. In the absence of this information, it remains questionable whether the mechanism is actually involved in the control of neurogenesis or whether the study revealed an interesting epiphenomenon.

Specific points:

While the authors state that they repeated some experiments in a second cell line, many of these are shown only qualitatively (Fig S2A) or not at all. Confirmation of the key findings in an independent ESC cell line is of high importance. Differentiation potential of individual human ESC lines is highly variable and therefore it is essential to establish the universality of the observations by probing several cell lines.

Reviewer #1 is absolutely right, and probing the role of CSDE1 in several ESC lines is needed to reach conclusions. We have now used 5 independent pluripotent stem cell lines (i.e., H9, H1, HUES6, HUES9 and iPSCs) and performed the following experiments to strengthen our conclusions:

- 1) We have assessed the decrease in CSDE1 protein levels during neural differentiation in two independent hESC lines as well as mouse ESCs (**Fig. 1b, 1j and Supplementary Fig. 3**).*
- 2) We have now quantified whether loss of CSDE1 induces spontaneous proliferation of PAX6-positive cells in a total of four independent hESC lines (i.e., H9, H1, HUES6 and HUES9). The text now says: "Because hESC lines can vary in their characteristics, we examined whether loss of CSDE1 induces proliferation of PAX6-positive cells in distinct lines. Indeed, knockdown of CSDE1 (**Supplementary Fig. 8b-d**) triggered spontaneous differentiation into OCT4-lacking cells that express high levels of PAX6 in all the lines tested (**Fig. 2h-j**)".*
- 3) In our first submission, we examined whether loss of CSDE1 accelerates neural differentiation upon neural induction treatment in both H9 and H1 hESCs. We have now performed experiments on HUES6 hESCs and obtained similar results. Unfortunately, we were not able to induce neural differentiation of the HUES9 line using our differentiation protocol. For this reason, we used a human iPSC line as an additional independent pluripotent stem cell line to perform these experiments. Thus, we have now examined neural differentiation under neural induction treatment on 4 pluripotent stem cell lines: H9, H1, HUES6 and iPSCs. The text now says: "We found that CSDE1 KD hESCs differentiate significantly faster into PAX6-positive cells than control hESCs (**Fig. 3a and Supplementary Fig. 9**). We obtained similar results with CSDE1^{-/-} hESCs (**Supplementary Fig. 10**). Likewise, other independent hESC lines*

as well as iPSCs also showed a faster neural differentiation on CSDE1 knockdown (**Supplementary Fig. 11**)”.

- 4) Moreover, we have now tested whether loss of CSDE1 increases FABP7 levels in 2 additional hESC lines as well as iPSCs. Altogether, we have observed increased levels of FABP7 in 4 hESC lines (H9, H1, HUES6 and HUES9) and 1 iPSC line upon CSDE1 knockdown. The text now says: “Western blot experiments confirmed increased protein levels of FABP7 upon knockdown of CSDE1 in four independent hESC lines as well as iPSCs, even when we did not observe changes in the early neural marker PAX6 (**Fig. 4b** and **Supplementary Fig. 17**)”.
- 5) In addition, we have found that loss of CSDE1 impairs the mRNA levels of multiple transcripts (e.g., DDIT4, SEMA4A, EPHB3, BMP4, FOXH1, ACVR2B) in several independent pluripotent stem cell lines (**Fig. 8a-d** and **Supplementary Data 7**).

Downregulation of CSDE1 is not clearly demonstrated and needs to be supported by more than qualitative western blot analysis. The loading is uneven and no quantifications has been provided. The levels of mRNA need to be quantified and expression in individual cells should be assessed by immunostaining. Furthermore, existing expression profiling data on neural differentiation of ESCs should be analyzed for changes in CSDE1 expression in these independent global datasets. Demonstrating the developmental regulation of CSDE1 is a key argument for its biological relevance.

We agree with Reviewer #1 that downregulation of CSDE1 protein needs to be supported by more than qualitative western blots. Besides the quantitative proteomics experiments shown in **Supplementary Table 1**, we have now analyzed existing quantitative protein expression data comparing hESCs with their NPC and neuronal counterparts (Noormohammadi et al, Nat. Comm. 2016). In support of our results, these data indicate that hESCs exhibit increased levels of CSDE1 when compared with NPCs and neurons. These analyses are now shown in **Fig. 1a** and discussed in the text: “First, we examined CSDE1 protein levels using available quantitative proteomics data comparing hESCs with their differentiated neural progenitor cell (NPC) and neuronal counterparts³⁴ (**Fig. 1a**). Notably, hESCs lost their high CSDE1 levels when differentiated into NPCs (**Fig. 1a**) as we confirmed by western blot analysis (**Fig. 1b** and **Supplementary Fig. 3**)”.

As indicated by Reviewer #1, we have now provided quantifications of the western blot experiments comparing ESCs with their differentiated counterparts (**Figure 1**). The graphs represent the relative percentage of CSDE1 protein levels (corrected for loading control) to ESCs. This was an excellent suggestion by Reviewer #1 that clearly shows the differences in CSDE1 amounts between ESCs and their differentiated counterparts. To further assess the decrease in CSDE1 levels during neural differentiation, we have now performed western blots (followed by quantification) in an independent hESC line as well as mouse ESCs.

Following Reviewer #1’s comments, we have now performed immunostaining experiments to assess the levels of CSDE1. First, we validate the specificity of the signal by comparing control hESCs with CSDE1 KD hESCs (please see figure below). Then, we compared hESCs with NPC cultures (after 2 weeks of neural induction treatment) (**Fig. 1e**). In addition, we performed immunostaining of cell cultures at early stages of neural differentiation treatment (2 days) to compare hESCs with differentiated cells in the same culture (**Fig. 1f**). The text now says: “To assess the levels of CSDE1 in individual cells, we performed immunocytochemistry experiments. In hESCs, CSDE1 was mostly localized in the cytoplasm and, particularly, concentrated in the perinuclear region (**Fig. 1e**) as previously

reported in other cell types³⁵. When hESCs were differentiated into NPCs, the expression of CSDE1 was downregulated (**Fig. 1e**). As a more formal test, we examined cell populations at early stages of the neural induction treatment to have a heterogeneous culture with hESCs and differentiated cells. Notably, cells expressing pluripotency markers exhibited higher amounts of CSDE1 when compared to differentiated cells (**Fig. 1f**)”.

Immunostaining experiments of control and CSDE1 KD hESCs with CSDE1 antibody. CSDE1 is mostly localized in the cytoplasm and, particularly, concentrated in the perinuclear region (as previously reported in other cell types). The intensity of the cytoplasmic signal is dramatically reduced on CSDE1 knockdown. We also observed nuclear signal with CSDE1 antibody in both control and CSDE1 KD hESCs. This nuclear signal could be unspecific, as it does not decrease on CSDE1 KD.

Regarding mRNA levels, we only examined protein levels in our first submission because CSDE1 is tightly regulated at the translation level (Schepens et al (EMBO J, 2007), Dormoy-Raclet et al (RNA Biol, 2005), Cornelis et al (Nucleic Acid Res, 2005)). Therefore, protein amounts may not correlate with mRNA levels. For instance, CSDE1 protein levels are decreased in Diamond-Blackfan anemia patient-derived erythroblasts via translational regulation whereas CSDE1 mRNA expression remains constant compared with control erythroblasts (Horos et al, Blood, 2012). Nevertheless, Reviewer #1 is right and it will be informative to add the analysis of mRNA levels in the manuscript. We have now quantified CSDE1 mRNA expression in hESCs and their differentiated counterparts (NPCs, neurons, mesoderm, cardiomyocytes, endoderm). We did not detect significant changes in CSDE1 mRNA levels during differentiation. These results are now shown in **Fig. 1g-i** and further commented in the Discussion section: “Given the strong phenotype observed upon CSDE1 knockdown, we focused on understanding the molecular mechanisms by which CSDE1 regulates hESC function. Interestingly, CSDE1 protein is highly abundant in hESCs and its levels decrease with differentiation into the distinct germ layers. However, the decrease in CSDE1 protein levels does not correlate with changes in mRNA expression. Thus, our results indicate that downregulation of CSDE1 protein expression during differentiation is regulated via translational or post-translational mechanisms. As other CSD-containing proteins, CSDE1 is mainly regulated at the post-transcriptional level²². CSDE1 negatively regulates its own translation by binding to an IRES in the 5' UTR of CSDE1 transcript. Besides CSDE1 itself, other proteins interact with the IRES of the CSDE1 transcript to either repress or enhance translation^{20,22}. For instance, polyprimidine tract-binding protein (PTB)

inhibits CSDE1 translation whereas hnRNP C1/C2 (HNRNPC) and specific ribosomal subunits stimulate CSDE1 protein expression^{20,36,24}. Since IRES regulators bind to CSDE1 transcript in a dynamic process²⁰, it will be fascinating to define modulators of CSDE1 levels during differentiation”.

The authors reported co-IP experiments to identify CSDE1 interactors. They list the identified proteins, but do not provide any data that would control for the quality and specificity of IPs. The primary data supporting the IP experiments should be added to supplemental information.

*We apologize for not providing the primary data for these experiments in our first submission. We have now included the primary data and analyses supporting the co-IP experiments (**Supplementary Data 2**). Likewise, we have now added the primary analysis data of all the proteomics and transcriptomics experiments (please see **Supplementary Data**). Moreover, we will deposit the raw data of these experiments in public repositories (i.e., PRIDE for proteomics data, SRA for sequencing data) and provide the corresponding accession codes in the manuscript to make the data publicly available.*

*The t-test Difference column in **Supplementary Table 2** and **Supplementary Data 2** represents the differences of the means (calculated from the log₂ of LFQ values) of the CSDE1 co-IP compared to FLAG control co-IP. The t-test Difference states the extent and direction of a change between CSDE1 and FLAG co-IP. As such, the quality and specificity of the co-IP experiments are supported by the observation that CSDE1 is the most enriched protein upon CSDE1 co-IP compared with FLAG co-IP (t-test Difference= 10.8). In addition, we have now included co-IP experiments with CSDE1 antibody (and FLAG antibody as a negative control) followed by western blot with CSDE1 as supporting data of the quality and specificity of the co-IP (**Supplementary Fig. 14a**).*

Finally CSDE1/UNR protein has been studied in detail in cancer. Previous study reported Vimentin as a target of CSDE1 and expression analysis as well as CLIP identification of CSDE1 targets has been performed. The authors need to properly reference the published papers and use existing data to perform comparative analysis. CSDE1 was also knocked out in mouse ES cells and no defect in ES cell proliferation has been noted. This deserves to be discussed in the manuscript.

As Reviewer’s #1 indicates, Dr. Gebauer’s laboratory has recently demonstrated a role of CSDE1 in melanoma invasion and metastasis. Moreover, they performed a comprehensive analysis to define direct targets of CSDE1 in melanoma. As the first systematic characterization of CSDE1 targets in mammalian cells, Dr. Gebauer’s findings have important implications for our work. For instance, they defined VIM as a direct target of CSDE1 in melanoma cells. We apologize for not citing this relevant study in our first submission. In fact, we have now discussed our results with Dr. Gebauer and she provided thoughtful advices for our project. Dr. Gebauer’s studies are now properly referenced and discussed in detail. Moreover, we performed comparative analysis with these data. The main changes related with this point are:

- *The introduction section now says: “Recently, a comprehensive study combining individual-nucleotide resolution crosslinking immunoprecipitation sequencing (iCLIP-seq), RNA sequencing and ribosome profiling unveiled CSDE1 targets in human*

melanoma¹⁸. In these cells, CSDE1 protein expression is often increased and regulates the levels of pro-oncogenic factors such as vimentin (VIM) or RAC1 as well as tumor suppressors (e.g., PTEN)¹⁸. Interestingly, this study also demonstrated that CSDE1 binds to mRNAs encoding regulatory proteins involved in development and neuron projection guidance in melanoma cells¹⁸.

- Results section: “Then, we performed quantitative proteomics to analyse their proteome. Besides CSDE1 levels, quantitative proteomics analysis revealed that other 20 proteins (out of 4435 quantified proteins in all the samples) are significantly changed in both independent CSDE1 shRNA hESC lines (**Supplementary Table 3 and Supplementary Data 3**). Among them, 10 proteins were up-regulated (e.g., FABP7 and VIM) whereas the others were down-regulated (e.g., THUMPD3). The transcripts of 6 significantly changed proteins (VIM, DNM1, SMARCC2, CSDE1, EIF4A2 and ANXA2) have been previously identified as direct RNA targets of CSDE1 in human melanoma cells¹⁸”.
- We have used the CSDE binding motif defined in Dr. Gebauer’s study to identify predicted CSDE1 targets in our RNA sequencing experiments (**Fig. 7**).
- We have also examined distinct direct targets of CSDE1 in melanoma cells as controls for our RIP experiments. The text now says: “To determine whether these transcripts are direct CSDE1 targets, we performed RIP experiments. As a positive control, we examined DDIT4, CDH2 and TMEM64 mRNAs that have been previously reported to interact with CSDE1 protein in melanoma¹⁸”.
- We have performed comparative analysis with the existing data of CSDE1 targets in melanoma cells. In particular, we have now examined whether the GO biological processes (GOBP) enriched in our transcriptomics analysis of CSDE1 KD hESCs overlap with the defined CSDE1 targets in melanoma cells. We found overlap in GOBPs such as extracellular matrix organization, organ development, anatomical structure morphogenesis and neuron projection guidance (as shown in **Supplementary Data 6**). The discussion section now says: “GOBP analysis of transcripts impaired on CSDE1 KD revealed a strong enrichment for regulators of extracellular matrix organization, development, neuron differentiation and neuron projection guidance. Interestingly, the direct CSDE1 targets identified in melanoma cells¹⁸ are also enriched for factors with a role in extracellular matrix organization, organ development, anatomical structure morphogenesis and neuron projection guidance (**Supplementary Data 6**).”
- In the discussion section, we have now extensively discussed the effects on VIM triggered by loss of CSDE1 in hESCs compared with melanoma.

Following Reviewer #1’s comments, we have now discussed the findings reported in Dormoy-Raclet et al (Oncogene, 2007) indicating that loss of CSDE1 does not affect proliferation of mouse ESCs. In addition, we have now examined the proliferation rates of hESCs on CSDE1 knockdown. The text now says: “Although loss of CSDE1 had no effect on hESC proliferation (**Supplementary Fig. 5**) as previously observed in mESCs³⁷(...)”.

Reviewer #2 (Neurogenesis expert):

Lee et al explore the function of CSDE1, an RNA binding protein, in the regulation of hESC self-renewal and neuronal differentiation. They test knockdown of CSDE1 in ES cells and suggest that this results in an accelerated neural induction and neurogenesis; this is indicated by increased FABP7 and vimentin (a radial glia/neural stem cell marker). They

propose a functionally important role for both FABP7 (also known as BLBP) in control of neurogenesis and that CSDE1 controls the levels through a posttranscriptional mechanism. Gain of function of CSDE1 can block neuronal differentiation. This is a novel observation.

CSDE1 emerged as a candidate regulator of hESCs from an initial proteomics and shRNA knockdown screen. Figure 1 present the characterisation of CSDE1 protein and mRNA levels. The western is not particularly convincing that the levels in neurons are decreased, as the loading control is also much lower.

We agree with Reviewer #2 that the loading control is uneven in the western blot showed in Fig. 1b (hESCs vs neurons) and further characterization is needed to support our findings. We have now performed 2 more neuronal differentiation experiments and provided quantification data of the western blot analyses for a total of 5 independent differentiation assays (Fig. 1b). As suggested by Reviewer #1 (please see above), we have also analyzed existing quantitative proteomics data (Noormohammadi et al, Nat. Comm., 2016) comparing hESCs with their NPC and neuronal counterparts. In support of our results, these data indicate that the levels of CSDE1 are lower in NPCs and neurons compared with hESCs. These analyses are now shown in Fig. 1a and discussed in the text: "First, we examined CSDE1 protein levels using available quantitative proteomics data comparing hESCs with their differentiated neural progenitor cell (NPC) and neuronal counterparts³⁴ (Fig. 1a). Notably, hESCs lost their high CSDE1 levels when differentiated into NPCs (Fig. 1a) as we confirmed by western blot analysis (Fig. 1b and Supplementary Fig. 3)." Taken together, the data presented in Fig. 1 and related Supplementary Figures support a downregulation of CSDE1 protein levels during differentiation.

There are indications of increased neural induction, but they need to better confirm knockdown and rescue the phenotype to ensure this is not an off target effect; obviously CRISPR deletion is feasible and has become rather standard definitive proof of function. They only present data using two cell lines, so limited insights are gained. Overall I am not convinced, as much more in depth analysis of the phenotype is needed.

While there are some interesting preliminary observations, the characterisation of the shRNA knockdown ES cells is rather superficial and premature. I believe this is premature for publication and unfortunately is some way away from being a significant enough body work with broad enough interest for Nature Communications. There is also no attempt to really integrate this CSDE1 function with the vast body of known regulators that control ES cell neural induction.

We completely agree with Reviewer #2 and we have now performed multiple experiments to strengthen our conclusions.

As Reviewer #2 indicates, CRISPR deletion has become rather standard definitive proof of function. Thus, we generated CSDE1 knockout hESCs by CRISPR/Cas9 system. To ensure that the observed phenotypes are not caused by an off target effect we provide the following results:

- 1) We have now included experiments with an additional shRNA against the 3'UTR of CSDE1. Taken together, we have now provided experiments with two independent shRNAs against the coding region of CSDE1 as well as one shRNA to the 3'UTR region supporting a role of CSDE1 in preventing neural*

differentiation of hESCs. Moreover, knockout of CSDE1 in hESCs also induced their spontaneous differentiation into PAX6-positive cells. The text now says: “We tested three independent shRNAs to CSDE1 and obtained similar results (**Fig. 2e, f and Supplementary Fig. 7**). Likewise, knockout of CSDE1 (**Supplementary Fig. 8a**) also resulted in the proliferation of PAX6-positive cells (**Fig. 2g**).”

- 2) Furthermore, we have now confirmed that loss of CSDE1 accelerates neural differentiation upon neural induction treatment with a third independent CSDE1 shRNA as well as CSDE1^{-/-} hESCs (Please see **Supplementary Figs 9-10**)
- 3) We confirmed changes in FABP7, VIM and other potential targets of CSDE1 in CSDE1^{-/-} hESCs (**Fig. 8e and Supplementary Data 7**). The text now says: “We confirmed that CSDE1^{-/-} hESCs also exhibit alteration in the steady-state levels of these transcripts whereas the expression of the pluripotency marker OCT4 was similar compared to control hESCs (**Fig. 8e and Supplementary Data 7**)”.

As indicated by Reviewer #2, validation of our results in multiple lines is of high importance. Reviewer #1 also raised this concern (please see above). To address this point we have now performed experiments in 5 independent pluripotent stem cell lines (i.e., H9, H1, HUES6, HUES9 and iPSCs). In particular, we have performed the following experiments:

- 1) We have now quantified whether loss of CSDE1 induces spontaneous proliferation of PAX6-positive cells in a total of four independent hESC lines (i.e., H9, H1, HUES6 and HUES9). The text now says: “Because hESC lines can vary in their characteristics, we examined whether loss of CSDE1 induces proliferation of PAX6-positive cells in distinct lines. Indeed, knockdown of CSDE1 (**Supplementary Fig. 8b-d**) triggered spontaneous differentiation into OCT4-lacking cells that express high levels of PAX6 in all the lines tested (**Fig. 2h-j**)”.
- 2) In our first submission, we examined whether loss of CSDE1 accelerates neural differentiation upon neural induction treatment in H9 and H1 hESCs. We have now performed experiments on HUES6 hESCs and obtained similar results. Unfortunately, we were not able to induce neural differentiation of the HUES9 line using our differentiation protocol. For this reason, we used a human iPSC line as an additional independent pluripotent stem cell line to perform these experiments. Thus, we have now examined neural differentiation under neural induction treatment on 4 pluripotent stem cell lines: H9, H1, HUES6 and iPSCs. The text now says: “We found that CSDE1 KD hESCs differentiate significantly faster into PAX6-positive cells than control hESCs (**Fig. 3a and Supplementary Fig. 9**). We obtained similar results with CSDE1^{-/-} hESCs (**Supplementary Fig. 10**). Likewise, other independent hESC lines as well as iPSCs also showed a faster neural differentiation on CSDE1 knockdown (**Supplementary Fig. 11**)”.
- 3) Moreover, we have now tested whether loss of CSDE1 increases FABP7 levels in 2 additional hESC lines as well as iPSCs. Altogether, we have observed increased levels of FABP7 in 4 hESC lines (H9, H1, HUES6 and HUES9) and 1 iPSC line upon CSDE1 knockdown. The text now says: “Western blot experiments confirmed increased protein levels of FABP7 upon knockdown of CSDE1 in four independent hESC lines as well as iPSCs, even when we did not observe changes in the early neural marker PAX6 (**Fig. 4b and Supplementary Fig. 17**)”.
- 4) In addition, we have found that loss of CSDE1 impairs the steady-state mRNA levels of multiple neural regulatory factor (e.g., DDIT4, SEMA4A, EPHB3, BMP4,

FOXH1, ACVR2B) in several independent pluripotent stem cell lines (**Fig. 8a-d and Supplementary Data 7**).

Following Reviewer #2's comments, we have performed a more in depth analysis of the phenotype. Our new experiments integrate CSDE1 function with the vast body of known regulators that control neural induction from hESCs.

- 1) By RNA sequencing experiments, we have now observed that loss of CSDE1 induces changes in the levels of numerous regulators of hESC identity, neural commitment and neurogenesis. Taken together, our results indicate that CSDE1 prevents changes in core components of multiple regulatory nodes involved in neural differentiation. The text now says: "GO biological process (GOBP) term analysis indicated strong enrichment for genes involved in the regulation of organismal development, cell differentiation, neurogenesis (e.g., DDIT4, EPHA2, TRPC6, FZD2, FZD7) and neuron projection development/guidance (e.g., NEFM, DAB2IP, EPHB3, SEMA4A, SEMA6C, EFNB1) (**Fig. 7c-d and Supplementary Data 6**). Moreover, we observed GOBP enrichment for modulators of the WNT signaling pathway (e.g., APCDD1, TMEM64, FZD7, GPC4, CDH2), a regulatory node of pluripotency and neural differentiation^{51,52} (**Fig. 7c-d and Supplementary Data 6**). Besides WNT signaling regulators, we also found changes in core components of known regulatory nodes of hESC identity and neural differentiation⁵¹ such as the BMP signaling pathway (e.g., BMP4, BMPR1A, ROR2), the FGF receptor signaling pathway (i.e., FGFR2, FGFR3, FGFR4), the TGF-beta/SMAD binding signaling pathway (e.g., FOXH1 and the activin A receptor ACVR2B) and LIN28A. Moreover, we observed changes in the steady-state transcript levels of genes involved in insulin/insulin-like growth factor binding (e.g., INSR, IGFBP5). Despite these alterations in numerous transcript levels, we did not observe downregulation of the pluripotency factor NANOG in our transcriptome analysis (**Supplementary Data 5**)".
- 2) In addition, we validated these changes in multiple pluripotent stem cell lines (please see **Fig. 8**).
- 3) We confirmed CSDE1 binding to specific transcripts and examined how CSDE1 post-transcriptionally regulates these mRNAs (please see **Fig. 9**).
- 4) Given that our results indicate that CSDE1 prevents neural differentiation of hESCs, we have examined whether CSDE1 downregulation also affects differentiation into other cell lineages. Our previous results suggested that loss of CSDE1 reduced the induction of endoderm markers upon definitive endodermal differentiation. We have now observed that CSDE1 downregulation also negatively affects the induction of cardiac mesoderm and cardiomyocytes markers. Thus, these results support our conclusion that CSDE1 downregulation commits hESCs to a neuroectoderm fate. We have now included the cardiomyocyte differentiation experiments in the manuscript: "Since our results indicated that loss of CSDE1 in hESCs facilitates their neural differentiation, we assessed the potential of CSDE1 KD hESCs to differentiate into other cell types. Notably, knockdown of CSDE1 reduced the induction of endoderm markers upon definitive endodermal differentiation when compared with control cells (**Fig. 3e**). Similarly, CSDE1 KD hESCs exhibited a diminished induction of cardiac mesoderm and cardiomyocytes markers during their progressive differentiation into cardiomyocytes (**Supplementary Fig. 13**)". Moreover, we have further discussed these results in the Discussion section: "In addition, knockdown of CSDE1 negatively affects hESC differentiation into distinct cell types such as definitive endoderm and

cardiomyocytes. A potential role of CSDE1 in mesoderm development is supported by the phenotype reported in Csde1^{-/-} mouse embryos, which exhibit delayed heart maturation with defects in ventricular trabeculation and atrioventricular cushions⁶¹”.

Reviewer #3 (RNA binding proteins and mechanisms expert):

This paper by Lee et al. investigated the role of CSDE1, an RNA binding protein with multiple cold shock domains, in maintenance of hESC pluripotency. The authors found knockdown of CSDE1 led to the spontaneous differentiation of hESCs to neuronal lineage, but not astroglial lineage. The authors further showed that CSDE1 KD led to increase of FABP7 and VIM, two novel CSDE1 targets that have been implicated in neuronal development, at the mRNA and/or protein levels, probably through direct protein-RNA interaction. Overall, the paper is well written, and the results add our understanding of the contribution of RBPs in stem cell biology.

Specific comments:

A prior study (ref 35) investigated knockout of CSDE1 in mouse ESCs and found that loss of CSDE1 led to differentiation into the endoderm lineage, which is in contrast to this study. While the paper was cited in discussion, the discussion was somewhat vague and focused on expression of markers. It will be helpful to clearly describe the previous study in introduction. This will be helpful to understand why markers of different germ layers were examined. Potential factors contributing to the discrepancy can also be discussed.

We agree with Reviewer #3 that these important findings by Elatmani et al need to be highlighted and further discussed in the main text:

- *First, we have now briefly summarized these findings in the introduction. The text now says: “Given the versatile binding of CSDE1 to mRNA targets and other RBPs, CSDE1 coordinates multiple biological processes¹⁴. The complexity of this regulation underlies the ability of CSDE1 to modulate the same biological process in an opposing manner depending on the cell type and state²²⁻²⁴. CSDE1 can either promote or inhibit apoptosis²² as well as differentiation in a cell-type specific manner. For instance, CSDE1 promotes erythroblast differentiation²⁴ whereas it prevents the differentiation of mouse naive ESCs into extraembryonic primitive endoderm-like cells²³”.*
- *Then, we further discussed the mESC study in the context of our results. We believe that this facilitates the comparison between the two studies. In addition, we have made more clear the differences between the two studies by highlighting the lack of spontaneous induction of endodermal markers in hESCs upon CSDE1 downregulation (**Supplementary Fig. 6**). The text now says: “When we grew hESCs without removing differentiated colonies, flattened cells proliferated in CSDE1 KD lines resulting in decreased levels of pluripotency markers such as OCT4, NANOG and DPPA2 compared with control hESCs (**Fig. 2c, d**). In addition, we observed a significant decrease in the percentage of OCT4-positive cells in CSDE1 KD lines (**Fig. 2e, f**). Although loss of CSDE1 had no effect on hESC proliferation (**Supplementary Fig. 5**) as previously observed in mESCs³⁷, our results revealed profound differences between these cells regarding cell fate decisions triggered by CSDE1 downregulation. mESCs are in a more naïve state than hESCs and retain*

their ability to differentiate into primitive endoderm^{38,39}, an extraembryonic tissue that maintains the expression of pluripotency markers while inducing high levels of endodermal markers such as GATA6^{23,40}. Whereas it has been reported that loss of CSDE1 induces the proliferation of primitive endoderm in mESC cultures²³, we found that CSDE1 downregulation in hESCs results in differentiated cells that lose the expression of pluripotency markers (**Fig. 2c-f**). Moreover, we did not observe an induction of GATA6 and other endodermal markers in hESCs (**Supplementary Fig. 6**)”.

- Finally, we have now extensively discussed several hypotheses that could explain the differences between the phenotypes observed in hESCs and those reported in mESCs. The discussion now says: “Whereas downregulation of CSDE1 in hESCs induces their differentiation into PAX6-positive cells that lose the expression of pluripotency markers, this RBP has been shown to prevent differentiation of mESCs into primitive endoderm-like cells²³. In the developing mouse embryo, primitive endoderm specification occurs within the inner cell mass from the mid-blastocyst stage⁴⁰. This extraembryonic tissue is characterized by maintaining high levels of pluripotency markers (e.g., OCT4, NANOG) while expressing endoderm markers such as GATA6, GATA4 or AFP^{23,40}. In order to explain the differences between our findings and those reported in mESCs, it is important to note the distinct pluripotent states exhibited by mESCs and hESCs in vitro³⁹. When cultured in serum and leukaemia inhibitory factor (LIF), mESCs are in a naive state resembling the pluripotent state observed in the inner cell mass of the pre-implantation embryo³⁹. As such, mESC cultures are heterogeneous and consist of at least two morphologically indistinguishable cell types, resembling either primitive endoderm lineages or primed progenitors of the epiblast⁶². On the other hand, hESCs are markedly different in vitro from mESCs and exhibit a more primed state that resembles post-implantation embryonic configurations³⁹. Thus, hESCs are more similar to epiblast stem cells than mESCs and lack the ability to differentiate spontaneously into primitive endoderm. Moreover, CSDE1 can modulate different biological pathways with high versatility depending on its interaction with specific proteins and target transcripts in a dynamic process associated with the cell type and status²²⁻²⁴. Interestingly, CSDE1 destabilizes GATA6 mRNAs in mESCs²³. However, we did not observe changes in GATA6 levels or other endodermal markers in hESCs. Therefore, CSDE1 could regulate distinct targets depending on the pluripotent state. In addition, mESCs and hESCs present distinct culture requirements that may impinge upon CSDE1 regulation of specific transcripts. For instance, LIF activates the JAK-STAT3 pathway and is a key ingredient for culturing mESCs in the absence of feeder cells³⁹. On the contrary, hESCs and human iPSCs do not require LIF signaling whereas FGF2 and TGFβ1/Activin A are core signaling pathways to maintain their pluripotent state³⁹. Remarkably, we found that CSDE1 modulates key components of FGF2 and TGFβ1/Activin A pathways in hESCs. Finally, we cannot discard that the distinct phenotypes observed in hESCs and mESCs upon CSDE1 downregulation are associated to unknown genetic differences between species. For instance, 81.5% of the CSDE1 mRNA targets identified in human melanoma do not overlap with those targets identified in *D. melanogaster*, indicating profound differences between species¹⁸”.

The evidence of direct regulation of FABP7 and VIM by CSDE1 is moderate. Not to mention RIP is probably not the ideal assay to determine the direct protein-RNA interactions, as

compared to other methods such as CLIP or mutagenesis of specific sites, the fold enrichment by RIP is moderate (4-5 fold), especially given that all tested substrates showed some extent of enrichment (which one is the negative control?).

*Since OCT4 and NES were not significantly enriched in our RIP experiments, we included them as negative controls. However, Reviewer #3 is absolutely right and these transcripts show some extent of enrichment. We have now examined other mRNAs in the same samples (i.e., B-actin and the neuronal genes SYN and AADC). These transcripts do not show enrichment in RIP experiments with CSDE1 antibody compared to FLAG antibody. Thus, we have now included these transcripts in the graph as negative controls. We have also indicated the FLAG values (= 1) in the graph to make more clear the enrichment of FABP7 and VIM transcripts with CSDE1 antibody. In collaboration with Dr. Gebauer's laboratory, we have now validated that in vitro transcribed biotinylated VIM and FABP7 mRNAs pull down CSDE1 protein from hESC lysates (please see **Supplementary Fig. 19**). Moreover, it is important to note that VIM mRNA has been recently reported as a direct target of CSDE1 in melanoma by Dr. Gebauer's laboratory (Wurth et al, Cancer Cell). We have now extensively discussed these findings in comparison with our results.*

*Besides FABP7 and VIM, our RNA sequencing experiments indicated that CSDE1 impinges upon the levels of multiple central components of known regulatory nodes of hESC identity, neuroectoderm commitment and neuron differentiation (e.g., factors involved in the development and guidance of neuronal projections). The aforementioned study in melanoma cells provided a consensus CSDE1 binding motif. We have now scanned predicted CSDE1 targets in our RNA sequencing experiments using the CSDE1 binding motif as a query (please see **Fig. 7** and **Supplementary Data 4**). After we defined neural regulator factors changed in multiple pluripotent stem cells upon CSDE1 knockdown, we performed RIP experiments to examine whether CSDE1 binds to these transcripts. Our results indicate that not all of these transcripts interact with CSDE1, suggesting pro-neural secondary effects induced by CSDE1 knockdown. The text now says: "To determine whether these transcripts are direct CSDE1 targets, we performed RIP experiments. As a positive control, we examined DDIT4, CDH2 and TMEM64 mRNAs that have been previously reported to interact with CSDE1 protein in melanoma¹⁸. Besides these transcripts, we found that CSDE1 also binds SEMA4A, GPC4, BMP4, FOXH1, EPHB3 and APCDD1 mRNAs in hESCs (**Fig. 9a**). In contrast, we did not observe interaction of CSDE1 protein with FZD7, SEMA6C, ACVR2B or IGFBP5, suggesting significant secondary effects induced by CSDE1 knockdown (**Fig. 9a**)".*

The heterogeneity of the colonies upon CSDE1 KD complicates the interpretation of the results. This issue is noted, but it might worth to elaborate more in discussion.

Reviewer #3 is right and we apologize for not making more clear this point in our first submission. To facilitate the interpretation of our results we have re-written the manuscript:

- *In the results section, we have now made more clear how we maintained CSDE1 KD cultures to avoid the proliferation of differentiated cells. The text now says: "Typical undifferentiated hESCs grow in tightly packed, three-dimensional colonies without spaces between cells. However, CSDE1 KD hESC cultures contained not only undifferentiated colonies but also monolayer colonies formed by flattened and elongated cells with reduced cell contact (**Fig. 2a** and **Supplementary Fig. 2**). This morphology contrasts with the growth pattern of control hESCs and other CSD-containing proteins KD hESCs, in which cells grew essentially in dense, three-*

dimensional colonies (**Fig. 2a and Supplementary Figs 1-2**). Accordingly, CSDE1 KD hESCs exhibited a decrease in alkaline phosphatase (AP) staining (**Fig. 2b**). Moreover, knockdown of CSDE1 also resulted in spontaneous neuronal differentiation (**Fig. 2a**). Given the strong phenotype observed in CSDE1 KD hESCs, we further characterized these cells. To maintain CSDE1 KD hESC lines, we transferred undifferentiated colonies followed by daily monitoring to remove differentiated cells (**Supplementary Fig. 4**). When we grew hESCs without removing differentiated colonies, flattened cells proliferated in CSDE1 KD lines resulting in decreased levels of pluripotency markers such as OCT4, NANOG and DPPA2 compared with control hESCs (**Fig. 2c, d**). In addition, we observed a significant decrease in the percentage of OCT4-positive cells in CSDE1 KD lines (**Fig. 2e, f**)”.

- We have now added a supplementary figure at the beginning of the results section that shows no differences in the percentage of OCT4 and PAX6-positive cells when CSDE1 KD hESC cultures were maintained by transferring undifferentiated colonies followed by daily monitoring to remove differentiated cells (please see **Supplementary Fig. 4**).
- Moreover, we have now indicated in all the figure legends when the cells were cultured without removal of differentiated cells to quantify changes in spontaneous differentiation rates. Likewise, we have also indicated in the main text and all the figure legends when the experiments were performed in hESCs that do not exhibit morphological differences (e.g., proteomics and transcriptomics experiments).
- As suggested by Reviewer #3, we have discussed in more detail the reasons why we used undifferentiated colonies for our proteomic and transcriptomic experiments (and related western blot and qPCR validation experiments). The text now says: “To define the mechanisms by which CSDE1 regulates neural differentiation and its targets in hESCs, we proposed to use undifferentiated CSDE1 KD colonies. Hence, we transferred typical undifferentiated colonies followed by extensive monitoring to remove differentiated cells. By these means, we avoided the use of CSDE1 KD heterogeneous populations that contain cells with prominent morphological differences. These cells could present an advanced state of differentiation that may mask the original events triggered by loss of CSDE1. In our first approach, we performed quantitative proteomic analysis of undifferentiated CSDE1 KD hESCs to identify these events”.
- In **Fig. 3**, we induced differentiation into neural cells (and other lineages) of undifferentiated hESCs. We have now made more clear this point not only in the main text but also in the Fig. 3 and Supplementary Figures 9-11 legends. The text now says: “With the spontaneous differentiation of CSDE1 KD hESCs into PAX6-positive cells, we asked whether this RBP is involved in the regulation of neural differentiation. To test this hypothesis, we selected undifferentiated hESC colonies and induced differentiation into NPCs (**Fig. 3a**)”.

REVIEWERS' COMMENTS:

Reviewer #1 (Remarks to the Author):

The revised manuscript addresses the major issues. For full disclosure, the authors should mention in the text that they failed to detect activation of neural differentiation program in HUES9 cell line.

Reviewer #2 (Remarks to the Author):

The authors have done an excellent job of addressing the main concerns of both myself and other reviewers. The additional hES cell lines, better quantitation, GO analysis are all extremely welcome and support their original conclusions, yet provide a much more convincing dataset.

I would suggest publication and have no major concerns.

Reviewer #3 (Remarks to the Author):

The revised manuscript is much improved. I have two questions that need clarification.

1. Fig. 3a vs. Supp Fig. 10. There are dramatic differences in the proportion of Oct4+ and PAX6+ cell even for the WT/control, although the experiments were performed in the same cell line. The discrepancy is not discussed in the manuscript.

2. in the new RNA-seq data that was included in the revised manuscript, the authors did not describe the criteria used to detect differentially expressed genes with statistical significance.

Reviewer #1

The revised manuscript addresses the major issues. For full disclosure, the authors should mention in the text that they failed to detect activation of neural differentiation program in HUES9 cell line.

We have now mentioned in the text that we failed to induce neural differentiation of HUES9 hESCs. The text now says: "Following this protocol, we were able to induce neural differentiation of H9, H1, HUES6 hESCs as well as iPSCs. However, we were not able to induce neural differentiation of the HUES9 hESC line".

Reviewer #2

The authors have done an excellent job of addressing the main concerns of both myself and other reviewers. The additional hES cell lines, better quantitation, GO analysis are all extremely welcome and support their original conclusions, yet provide a much more convincing dataset.

I would suggest publication and have no major concerns.

Reviewer #3

The revised manuscript is much improved. I have two questions that need clarification.

1. Fig. 3a vs. Supp Fig. 10. There are dramatic differences in the proportion of Oct4+ and PAX6+ cell even for the WT/control, although the experiments were performed in the same cell line. The discrepancy is not discussed in the manuscript.

As Reviewer #3 indicates, we observed a slower differentiation rate into PAX6-positive cells in the new neural differentiation experiments showed in Supplementary Figure 10 compared with Figure 3a. Although neural induction was performed using H9 hESCs in both set of experiments, it is important to note that the control cells in Figure 3a express a non-targeting shRNA. Accordingly, these control cells were treated with puromycin every 3-4 days as CSDE1 shRNA H9 hESCs to select shRNA-expressing cells. On the contrary, both wild-type and CSDE1^{-/-} H9 hESCs were not maintained under regular puromycin treatments. This could contribute to the different rates observed in both neural induction experiments. As mentioned by Reviewer #3, it is important to discuss these differences in the manuscript. We have now mentioned this in the Figure legend of Supplementary Figure 10 (highlighted in red font).

We believe that we used the proper controls for each experiment (i.e., Non-targeting shRNA vs CSDE1 shRNA H9 hESCs and wild-type vs CSDE1^{-/-} H9 hESCs). Both experiments independently indicate a role of CSDE1 in modulating the neural differentiation ability of H9 hESCs. Since hESCs often show variability in their differentiation efficiency even between independent replicate experiments, our data using independent hESC lines were important to reach solid conclusions.

2. in the new RNA-seq data that was included in the revised manuscript, the authors did not describe the criteria used to detect differentially expressed genes with statistical significance.”

We apologize for not describing the criteria used to detect differentially expressed genes with statistical significance in our RNA-seq experiment. Differential gene expression analysis was performed with cuffdiff (version 2.2.1). Briefly, all genes at a FDR level below 0.05 were retained as significantly differentially expressed. We have now indicated (highlighted in red font) this criteria in figure legend 7, methods and the excel file containing the RNA-seq data (Supplementary Data 5).